# Project "Biodiversity MARE Tricase": A Species Inventory of the Coastal Area of Southeastern Salento (Ionian Sea, Italy)

Valerio Micaroni [1,2,*,†], Francesca Strano [1,2,*,†], Fabio Crocetta [3], Davide Di Franco [4], Stefano Piraino [5,6,7], Cinzia Gravili [5,7], Fabio Rindi [8], Marco Bertolino [9], Gabriele Costa [9,10], Joachim Langeneck [7], Marzia Bo [7,9], Federico Betti [9], Carlo Froglia [11], Adriana Giangrande [5], Francesco Tiralongo [12,13], Luisa Nicoletti [14], Pietro Medagli [5], Stefano Arzeni [5] and Ferdinando Boero [15,16,17]

1 School of Biological Sciences, Victoria University of Wellington, Wellington 6140, New Zealand
2 MARE Outpost, University of Salento, 73039 Tricase, Italy
3 Department of Integrative Marine Ecology (EMI), Stazione Zoologica Anton Dohrn, Villa Comunale, 80121 Naples, Italy
4 Institute for Ecology, Diversity and Evolution, Goethe University Frankfurt, 60325 Frankfurt am Main, Germany
5 Department of Biological and Environmental Sciences and Technologies-DiSTeBA, University of Salento, 73100 Lecce, Italy
6 Museo di Biologia Marina "Pietro Parenzan", 73010 Porto Cesareo, Italy
7 Consorzio Nazionale Interuniversitario per le Scienze del Mare (CoNISMa), U.L.R di Lecce, Department of Biological and Environmental Sciences and Technologies-DiSTeBA, University of Salento, 73100 Lecce, Italy
8 Department of Life and Environmental Sciences, Polytechnic University of Marche, 60131 Ancona, Italy
9 Department for the Earth, Environment and Life Sciences, University of Genoa, 16132 Genoa, Italy
10 Department of Earth and Environmental Sciences, University of Milano-Bicocca, 20126 Milan, Italy
11 Institute of Marine Sciences, National Research Council, 60125 Ancona, Italy
12 Ente Fauna Marina Mediterranea, Scientific Organization for Research and Conservation of Marine Biodiversity, 96012 Avola, Italy
13 Department of Biological, Geological and Environmental Sciences, University of Catania, 95124 Catania, Italy
14 Italian National Institute for Environmental Protection and Research, 00144 Rome, Italy
15 Istituto per lo Studio Degli Impatti Antropici e Sostenibilità in Ambiente Marino (IAS), Consiglio Nazionale delle Ricerche, Italy Stazione Zoologica Anton Dohrn, Villa Comunale, 80121 Naples, Italy
16 Stazione Zoologica Anton Dohrn, Villa Comunale, 80121 Naples, Italy
17 Consorzio Nazionale Interuniversitario per le Scienze del Mare (CoNISMa), 00198 Rome, Italy
* Correspondence: micaroni.valerio@gmail.com (V.M.); francesca.strano.mare@gmail.com (F.S.)
† These authors contributed equally to this work.

**Abstract:** Biodiversity is a broad concept that encompasses the diversity of nature, from the genetic to the habitat scale, and ensures the proper functioning of ecosystems. The Mediterranean Sea, one of the world's most biodiverse marine basins, faces major threats, such as overexploitation of resources, pollution and climate change. Here we provide the first multi-taxa inventory of marine organisms and coastal terrestrial flora recorded in southeastern Salento (Ionian Sea, Italy), realized during the project "Biodiversity MARE Tricase", which provided the first baseline of species living in the area. Sampling was carried out by SCUBA and free diving, fishing gears, and citizen science from 0 to 70 m. Overall, 697 taxa were found between March 2016 and October 2017, 94% of which were identified to the species level. Of these, 19 taxa represented new records for the Ionian Sea (36 additional new records had been reported in previous publications on specific groups, namely Porifera and Mollusca Heterobranchia), and two findings represented the easternmost records in the Mediterranean Sea (*Helicosalpa virgula* and *Lampea pancerina*). For eight other taxa, our findings represented the only locality in the Ionian Sea, besides the Straits of Messina. In addition to the species list, phenological events (e.g., blooms, presence of reproductive traits and behaviour) were also reported, with a focus on gelatinous plankton. Our results reveal that even for a relatively well-known area, current biodiversity knowledge may still be limited, and targeted investigations are needed to fill the gaps. Further research is needed to understand the distribution and temporal trends of Mediterranean biodiversity and to provide baseline data to identify ongoing and future changes.

**Keywords:** species list; checklist; benthos; gelatinous plankton; temperate mesophotic ecosystems; citizen science

---

## 1. Introduction

Biodiversity is the most valuable resource on Earth [1]. In marine ecosystems, increased diversity of structural elements and functional traits are correlated with optimal resource use, which consequently increases and stabilizes energy and matter flows [2]. Conversely, biodiversity loss is associated with decreased productivity, recovery potential, and ecosystem stability [3]. Marine biodiversity provides key ecosystem goods and services such as food, raw materials, pharmaceuticals, genetic information, climate regulation, tourism, recreation, and many others [4]. Recent estimates (2011) of global ecosystem services ranged between US$125 and US$145 trillion/year, with coastal marine ecosystems as the most valuable biome, accounting for 22–27% of the total ecosystem services [5]. Nowadays, the importance of marine biodiversity is recognised by the scientific community, decision makers and citizens [6]. For instance, the European Union chose "Biodiversity is maintained" as the first descriptor of good environmental status in its Marine Strategy Framework Directive (2008/56/EC).

The Mediterranean Sea is a biodiversity hotspot, hosting more than 17,000 species, 20% of which are endemic [7]. This is particularly relevant considering that the Mediterranean Sea covers only 0.82% of the world's ocean surface area [8]. Humans have benefited from Mediterranean marine ecosystems and their services for millennia, profoundly affecting their biodiversity [9]. Historically, coastal urbanisation pressure, habitat degradation and overexploitation of resources have been the main drivers of biodiversity changes [10]. However, following the industrial revolution and the demographic explosion of the last century, new threats have emerged, and these changes have dramatically accelerated [9]. Recent anthropogenic threats include industrial pollution, eutrophication, microplastics, and non-indigenous species [7]. Furthermore, climate change is altering the distribution, phenology and physiology of many animals, algae, and plants [11–15]. On the one hand, the gradual rise in global temperatures is promoting the spread of warm-water organisms in the Mediterranean, leading to the so-called process of "tropicalization" [16]. On the other hand, extreme thermal events, also called marine heatwaves, have already caused several mass mortalities in the Mediterranean Sea and will likely worsen in the future [13,17–19]. Although the Mediterranean basin is one of the most studied seas in the world, there are still important gaps in basic knowledge, such as taxonomy, distribution, abundance, and temporal trends of occurrence of most taxa [7,20]. There are many regional-scale databases, but the knowledge of local and small-scale biodiversity is lacking [21]. This gap hinders the reliable estimates of current changes in biodiversity patterns and the prediction of future scenarios. Hence, there is an urgent need for biodiversity data at a high taxonomic resolution, including species inventories and long-term data series [22,23].

The MARE Outpost was established in Tricase (southeastern Salento, Apulia, Italy) to monitor the local coastal and marine biodiversity and to investigate the phenology and behaviour of marine organisms [24,25]. The remarkable biological uniqueness of this stretch of coastline has been recognized since the 1970s [26,27] and was confirmed by recent findings [28]. Due to its naturalistic importance, Tricase Porto is included in a forthcoming Marine Protected Area that will extend from Otranto to Leuca [29]. Following the institution of the MARE Outpost, the project "Biodiversity MARE Tricase" was initiated to inventory the coastal and marine biodiversity of the area and promote public environmental knowledge and awareness [24].

In the last 50 years, several species lists have been published for the area, but they were all almost exclusively limited to specific taxonomic groups (Mollusca Heterobranchia, [30,31]; shell-bearing Mollusca [32]; Porifera [26,33,34]; Hydrozoa [35]; Polychaeta [36]; Macroalgae [37]), or specific habitats (e.g, caves, [38,39]).



The main aim of this work was to increase current knowledge on the biodiversity of the Central Mediterranean Sea, providing the first multi-taxa species inventory of the coastal area of the southeastern Salento peninsula (Ionian Sea). In addition to the species list, data on depth distribution and phenology are provided, with a special focus on gelatinous plankton.

## 2. Materials and Methods

### 2.1. Study Sites and Sampling Methods

Samplings were carried out on the southeastern coast of the Salento Peninsula, in the Otranto Channel (Ionian Sea), between March 2016 and October 2017. Samplings were mostly conducted in an area of ~33 km$^2$ (Figure 1), having as the northern boundary the latitude 39°59′30.7″ N, the southern boundary the latitude 39°54′13.5″ N, the western boundary the coastline, and the eastern boundary a bathymetric depth of 70 m (the depth limit for local fishermen). In addition to the area described above, four additional sites were investigated: Acquaviva's cave, Bortone's cave, Ciolo's cave and the Castro's mussel farm (Figure 1). The specific sampling location with coordinates (available for 63% of the specimens recorded) is provided in Supplementary Material.

The sampling area is characterised by different substrate and habitat types. A steep rocky substrate, mainly consisting of limestone bedrock, is found from the surface to about 18 m depth. At higher depths, the slope decreases and patches of coralligenous banks (up to 3 m high) alternate with sand and mud down to about 70 m [27]. The area is also characterised by many karst freshwater springs and marine karst caves [40].

Sampling activities focused on the following ecological categories: hard and soft bottom phyto- and zoobenthos, gelatinous zooplankton, nekton, and terrestrial coastal vegetation. Sampling was carried out by free diving to sample benthic organisms, gelatinous plankton, and nekton from 0 to 20 m; by SCUBA diving, to sample benthos and nekton between 0 and 45 m; with fishing gear (set nylon trammel nets, set monofilament gillnets, pole and line and long line) to sample nekton, benthos, and substrates from both catch and bycatch between 0 and 70 m; by citizen scientists (i.e., locals, bathers, divers, and fishermen) to sample between the intertidal and 20 m.

Sampling in SCUBA and free diving was mainly carried out using a visual collection technique [41]. Compared to traditional sampling methods (quadrats, transects, substrate scraping), this visually oriented technique is equivalent in terms of species yields, but it is advantageous in terms of processing time and effort [12]. Samples of specimens of visible size (>5 mm) were collected physically and/or photographically. In addition, pieces of rocky and biogenic substrate were collected for subsequent screening and sorting in the laboratory under a dissecting microscope (Leica MZ6) to search for sessile and vagile organisms. In target stations, soft sediment (detritus) was collected to search for vagile organisms (mainly molluscs).

Sampling with artisanal and recreational fishing gear involved the visually oriented collection of visible catch and bycatch, and the screening of pieces of rocky and biogenic substrate that were collected by the nets for subsequent screening and sorting in the laboratory.

Citizen scientists sampling involved the physical and/or photographical collection of specimens by members of the public, such as locals, tourists, bathers, spear fishers, diving shops, and local fish shops.

In situ and laboratory photos were taken using cameras: Canon Power-Shot D30, Sony Rx100, Nikon AW130, and Sony A7 II. Overall, 134 sampling sessions were carried out. For most of the collected organisms, phenological traits (i.e., seasonality, presence of reproductive structures, mating behaviour) were also noted by stereoscope/light microscope observation and recorded.

### 2.2. Organism Preservation and Species Identification

Specimens that could not be identified in situ or from photographic material were sampled and brought to the field station, where they were photographed in vivo, and fixed

in EtOH and/or formalin 10% as reference material, following the protocol for "Invertebrate Specimen Processing Procedures: Methods of Fixation and Preservation" of the Department of Invertebrate Zoology of the Smithsonian National Museum of Natural History.

Preliminary identification of taxa was made using available taxonomic guides [42–57] together with specific literature and original descriptions. Successively, samples and photographic materials were analysed by experienced taxonomists for accurate species identification. Nomenclature follows the World Register of Marine Species [58], and AlgaeBase [59], although authorities are only reported in tables (see below).

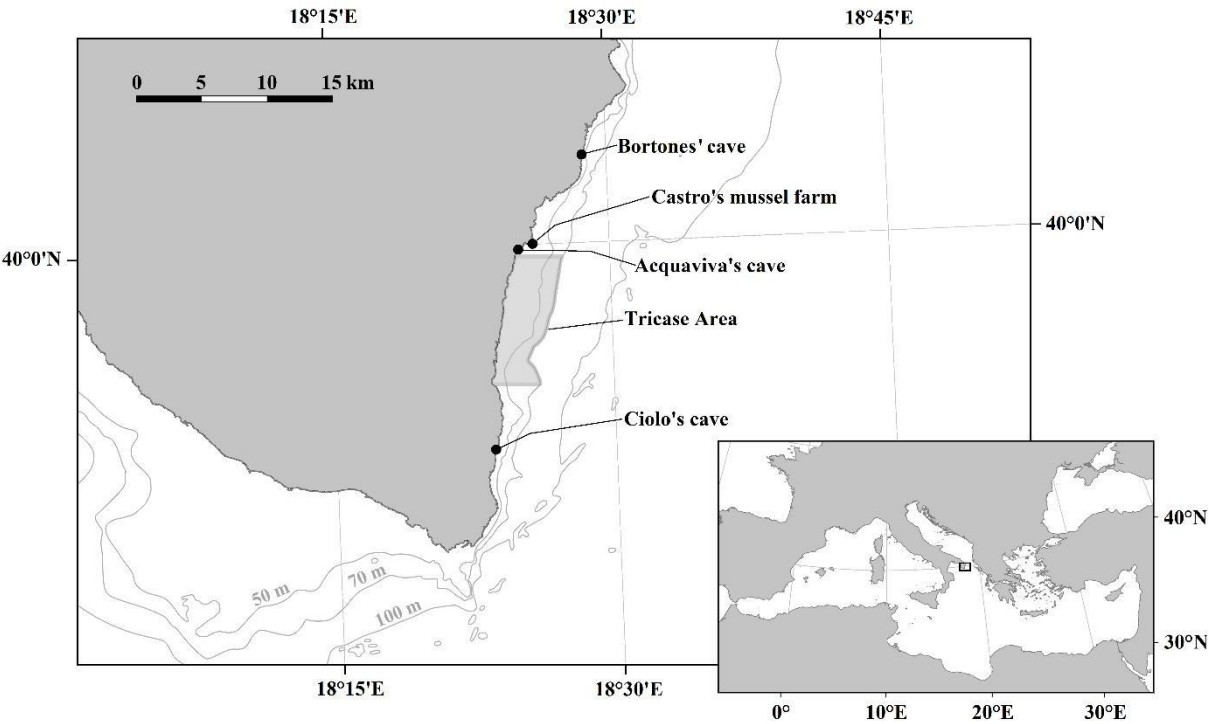

**Figure 1.** Map of the southeastern tip of the Salento Peninsula (Italy). The Tricase Area is highlighted in gray; black dots indicate the additional study sites. In the bottom right corner: the location of the study area in the Mediterranean Sea. After Micaroni et al. [49].

## 3. Results

### 3.1. General Description of Marine Biodiversity in the Area

The study area was characterized by rocky substrates from the surface to about 18 m. Below, the slope decreased, and rocky outcrops alternated with sand. In the supralittoral zone, *Melarhaphe neritoides* (Gastropoda) and *Ligia italica* (Isopoda) were abundant everywhere, along with algae such as *Cladophora dalmatica* and *Cladophora laetevirens* (Figure 2a). Algae such as *Ellisolandia elongata* and molluscs such as *Lepidochitona caprearum* and *Patella rustica* were common in the intertidal zone. The fucalean brown alga *Ericaria amentacea* was common in the infralittoral fringe (0–0.5 m), although it formed dense and continuous belts in a limited number of places. Algae such as *Dictyota dichotoma* (very common and abundant everywhere from 0 to 30 m), *Jania rubens* and species of the *Laurencia* complex (*Laurencia* cfr. *obtusa*, *Laurencia glandulifera*, *Palisada perforata*) were well represented in the upper infralittoral zone, from 0 to 3 m (Figure 2b). Then, from 3 to 13 m, the contribution of the *Laurencia* complex decreased, and other algae became abundant, such as *Padina pavonica*, *Halimeda tuna* and *Flabellia petiolata* (Figure 2c). Together with these algae, the sponges *Sarcotragus spinosulus* and *Chondrosia reniformis* were very common between 1 and 7 m.

At 14–18 m, coralligenous formations started. Calcareous algae such as *Peyssonnelia heteromorpha*, *Peyssonnelia rosa-marina* and *Peyssonnelia squamaria*, were very abundant in this habitat. Here, widespread sponges were *Petrosia (Petrosia) ficiformis*, *Agelas oroides*, *Axinella*

*cannabina*, *A. damicornis*, *A. polypoides*, *A. verrucosa* and *Acanthella acuta*. Very abundant were the bryozoans *Schizoretepora serratimargo*, which formed large (>100 cm) bioconstructions in the upper mesophotic zone (40–60 m), *Schizobrachiella sanguinea*, and *Adeonella pallasii*. Patches of *Posidonia oceanica* were found from 15–18 m, interspersed with coralligenous formations (Figure 2d).

The Rio area was characterized by the presence of wastewater treatment plant discharges, so it was likely more eutrophic than its surroundings and hosted a different algal assemblage. In particular, *Colpomenia sinuosa*, *Gelidium spinosum* and *Ellisolandia elongata* were abundant (Figure 2e). In addition, *Ericaria amentacea* formed very dense belts in spring, with much longer fronds than in other areas.

The copious karst caves scattered around the coast hosted distinctive communities and high biodiversity. Very common species were *Paractaea monodi*, *Herbstia condyliata* (Brachyura), *Polycyathus muellerae* (Anthozoa) and *Prostheceraeus giesbrechtii* (Polycladida) feeding on *Pycnoclavella* sp. (Ascidiacea). In addition, a high diversity of encrusting and massive sponges such as *Aplysina cavernicola*, *Fasciospongia cavernosa*, and species of the genera *Haliclona*, *Petrosia*, and *Plakina* were found (Figure 2f).

*3.2. Overall Species Inventory*

Overall, we sampled and recorded 1032 specimens belonging to 697 taxa, 655 of which were identified at the species level (94% of taxa), 11 at the species complex level, 28 at the genus level, and 3 at the family level—538 taxa are firstly reported here, while 159 were reported in Micaroni et al. [30] and Costa et al. [33]. The complete list of species is reported in Table 1 and Supplementary Material Table S1; the list of recorded and identified specimens with additional metadata (depth, site, coordinates, substrate, sampling method, and notes) is reported in Table S2.

The most diverse phyla were Mollusca (144), Porifera (112), Chordata (108), Cnidaria (85), Arthropoda (62), Rhodophyta (52) and Annelida (33) (Figure 3a). Among Mollusca, the most diverse classes were Gastropoda (107) and Bivalvia (30), while the most diverse orders were Nudibranchia (29), Littorinimorpha (19), and Neogastropoda (19). Most Chordata belonged to the classes Actinopterygii (89, mostly Perciformes: 64) and Ascidiacea (9). Regarding cnidarians, most taxa belonged to the classes Hydrozoa (49) and Anthozoa (31) (Figure 3b).

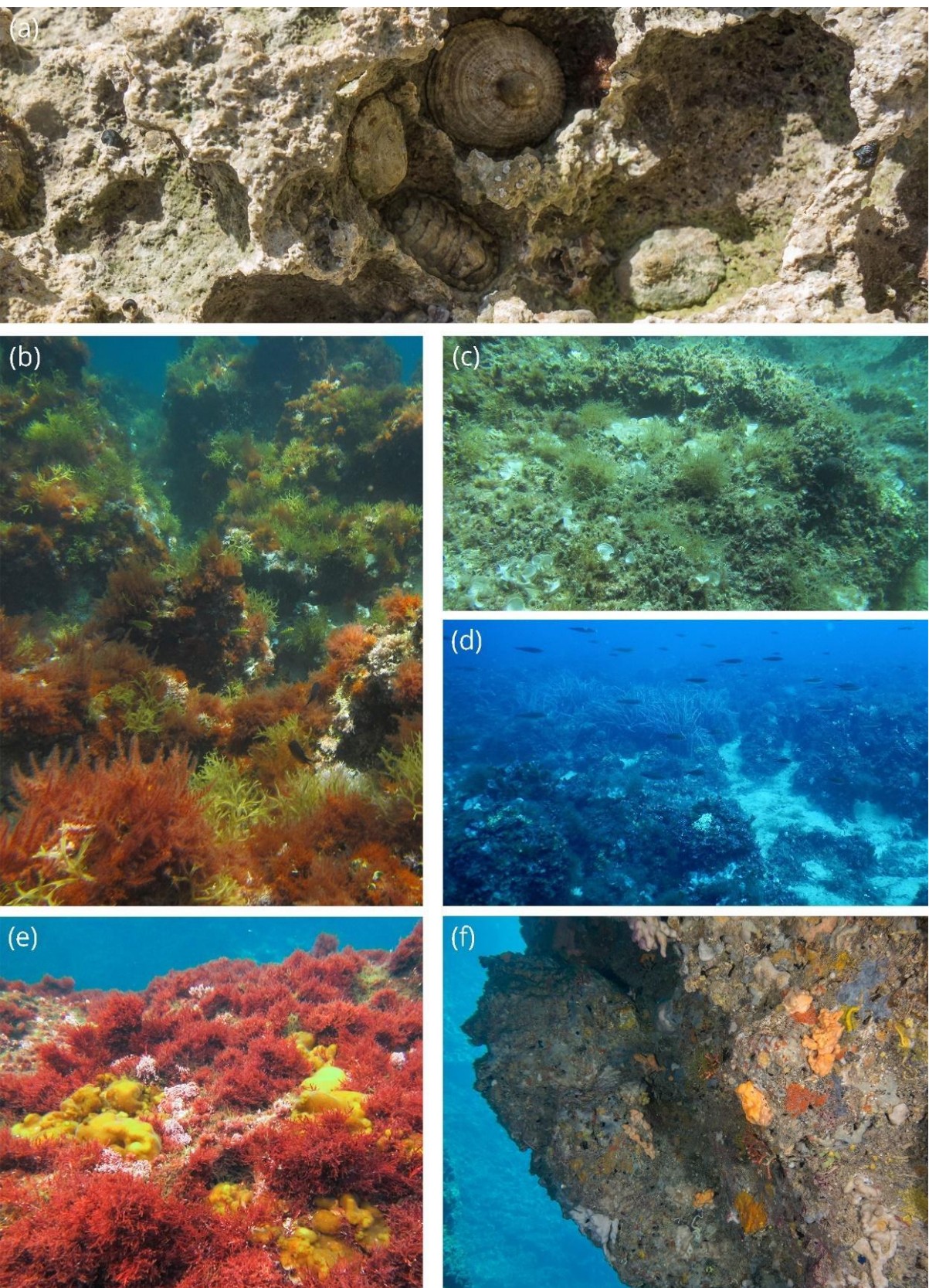

**Figure 2.** Marine habitats found in the coastal area of Tricase. (**a**) Intertidal zone, (**b**) upper infralittoral (2–4 m), (**c**) lower infralittoral (10–13 m), (**d**) coralligenous formation with *Posidonia oceanica* patches (17–22 m), (**e**) eutrophic infralittoral, (**f**) cave environment.

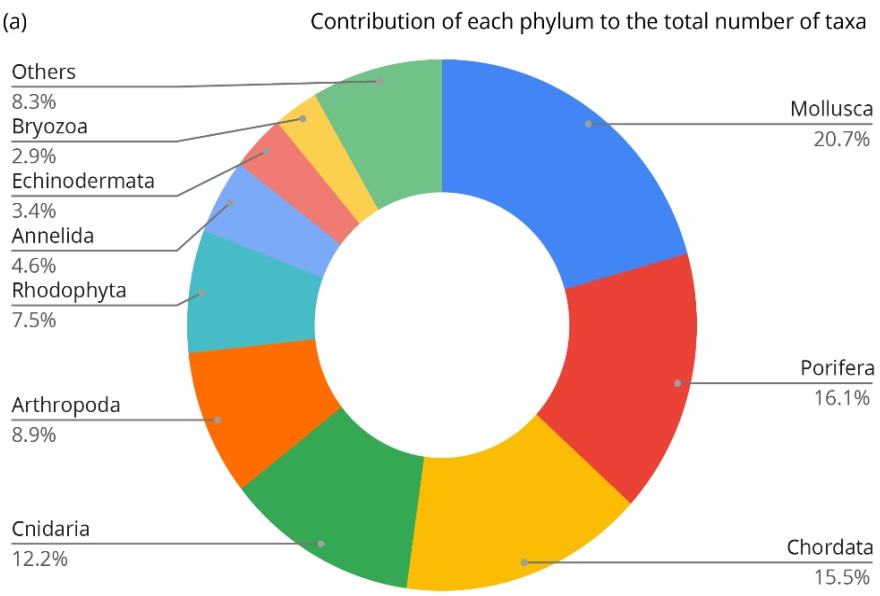

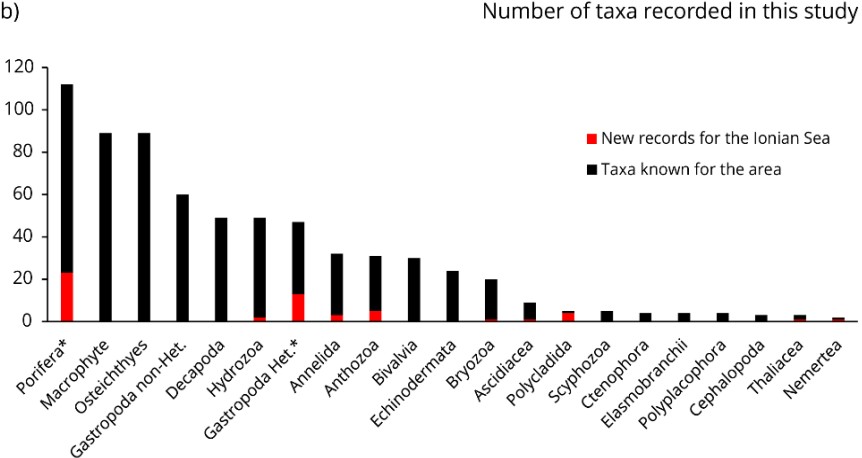

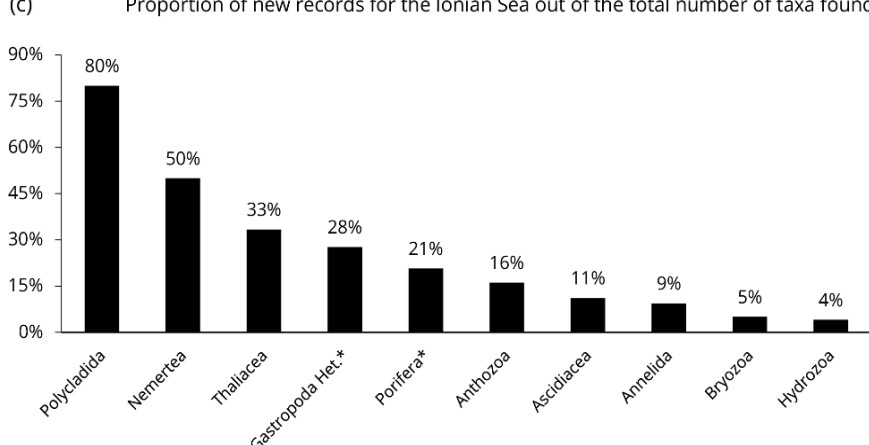

**Figure 3.** General description of the taxonomic composition and new records found by this study. (**a**) Relative contribution of each phylum to the total number of taxa. (**b**) Total number of taxa and new records (in red) for each major taxonomic group. (**c**) Percentage proportion of new records for the Ionian Sea out of the total number of taxa found in this study. Het., Heterobranchia. * indicates taxonomic groups that have been already included in previous publications [30,33].

Of all identified taxa, 55 represented new records for the Ionian Sea (36 of which were reported in Micaroni et al. [30] and Costa et al. [33]) (Table 1; Figures 3b, 4 and 5). Most of the new records belonged to the phyla Porifera (23, Costa et al., 2019), Mollusca (13), Cnidaria (7, of which 5 belonged to the class Anthozoa and 2 to the class Hydrozoa), Platyhelminthes (4, all from the order Polycladida), and Annelida (3). For other 8 taxa, our finding represented the only location in the Ionian Sea other than the Strait of Messina, considered by many authors to be a different biogeographical sector [16,60–62] (Table 1). Furthermore, records of *Helicosalpa virgula* (Chordata: Thaliacea) and *Lampea pancerina* (Ctenophora: Tentaculata) account for the easternmost sightings in the Mediterranean Sea (Table 1).

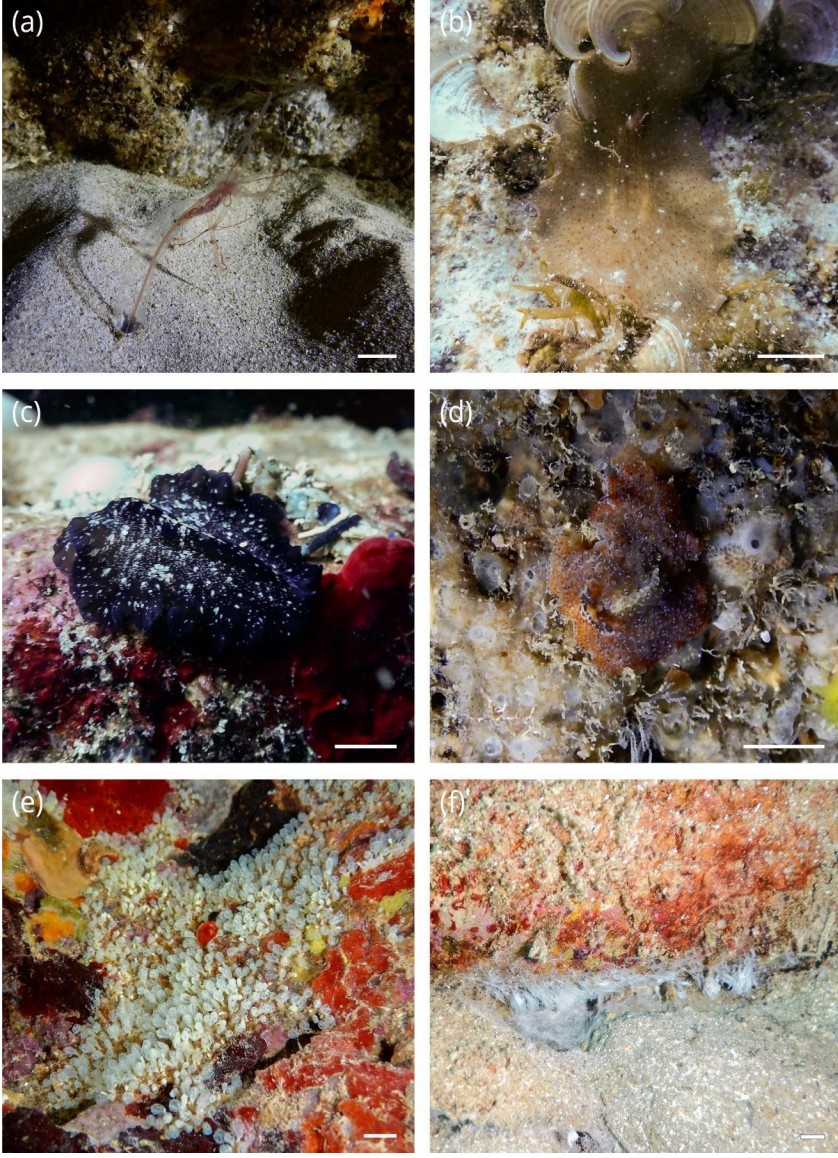

**Figure 4.** Photos of selected new records in the field. (**a**) *Halcampoides purpureus* on sand in Matrona Cave at 4 m; (**b**) *Planocera ceratommata* under a boulder on a rocky reef at 8 m; (**c**) *Pseudoceros maximus-type A* under a boulder on a rocky reef at 15 m; (**d**) *Thysanozoon brocchii* on a vertical wall in the Bortone's cave at 8 m; (**e**) *Pycnoclavella* sp. at the entrance of Matrona Cave, 15 m; (**f**) colonies of *Zoothamnium niveum* at the edge between a rocky cliff and sand, 10 m. The scale bars correspond to 1 cm.

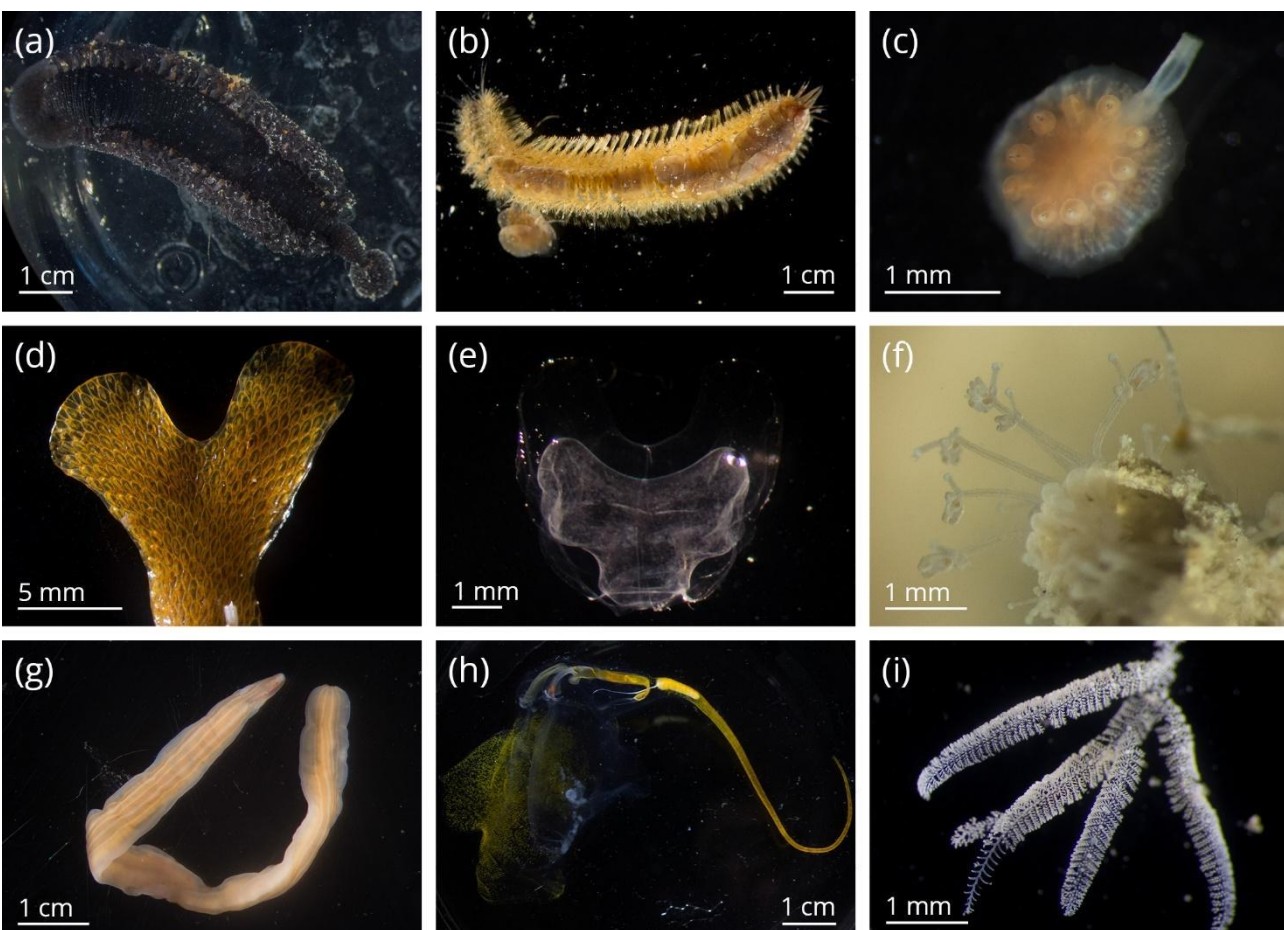

**Figure 5.** Photos of selected new records under a dissection microscope, in vivo. (**a**) *Branchellion torpedinis*; (**b**) *Harmothoe pagenstecheri*; (**c**) *Myzostoma glabrum*; (**d**) *Chartella* ind; (**e**) nectophore of *Halistemma rubrum*; (**f**) *Proboscidactyla ornata*; (**g**) *Gibsonnemertes spectabilis*; (**h**) *Helicosalpa virgula*; (**i**) *Zoothamnium niveum*.

Polycladida (Platyhelminthes) was the taxonomic group with a higher proportion of new records on the total number of taxa, with 80% of the species found being new records for the Ionian Sea (4 out of 5 taxa), followed by Nemertea (50%, 1 out of 2), Thaliacea (33%, 1 out of 3), Heterobranchia (28%, 13 out of 47), Porifera (21%, 23 out of 112), and Anthozoa (16%, 5 out of 31) (Figure 3c). Similar results were found comparing the proportion of new records in this study to the total number of taxa found in the Italian Ionian Sea by the checklist of the Italian marine fauna [60,61], with Polycladida (Platyhelminthes), Nemertea, Anthozoa, Porifera, Thaliacea, and Heterobranchia being the taxa with the highest proportion of new records (Figure 6a).

The majority of taxa (48.8%) were recorded between 0 and 10 m, whereas 29.2% of taxa were sampled between 11 and 25 m and 21.9% from 26 to 70 m. The highest percentage of new records (10%) has been reported for the depth range 11–25 m (Figure 7).

For 63 of the taxa found, phenology traits were also reported. These included the presence of reproductive structures (31), the occurrence of seasonal organisms (25), and instances of reproductive behaviour (6) (Tables 1 and S1).

Overall, we found 9 non-indigenous species (NIS), including two polychaetes (*Lysidice collaris* and *Hydroides elegans*), two crustaceans (*Callinectes sapidus* and *Percnon gibbesi*), two hydroids (*Clytia linearis* and *Eudendrium merulum*), and one ascidian, macroalga and ctenophore (*Botrylloides niger*, *Caulerpa cylindracea* and *Mnemiopsis leidyi*) (Table 1). None of these represent a new record for the area.

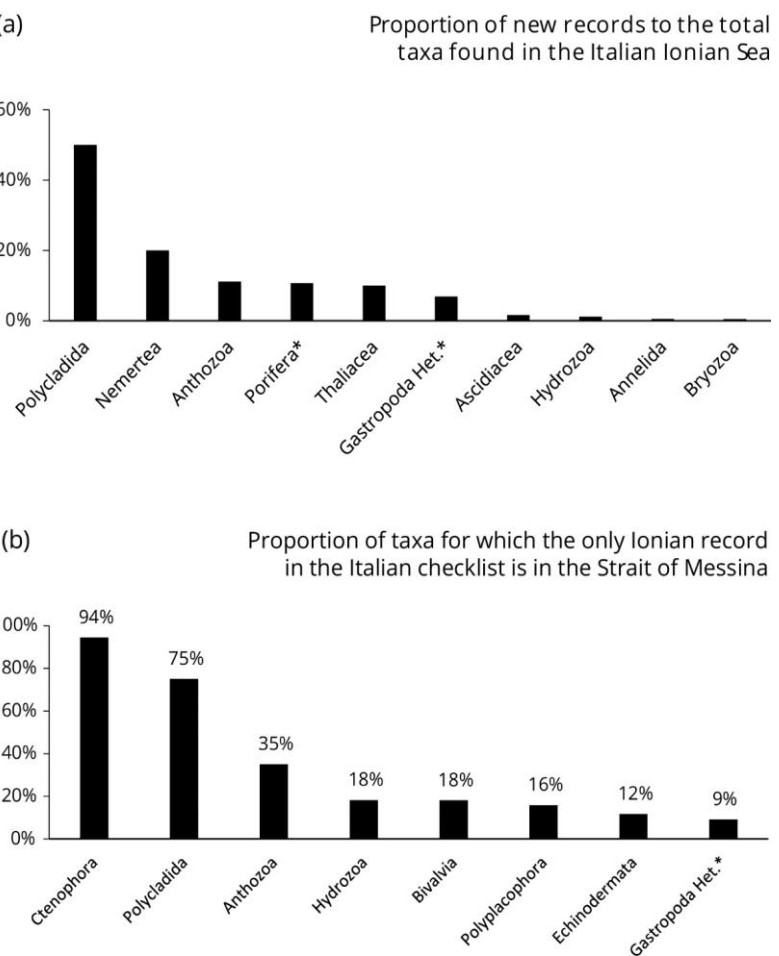

**Figure 6.** Comparison with the check list of the Italian marine fauna [60,61]. (**a**) Percentage proportion of new records found by this study to the total taxa reported in the Italian Ionian Sea by Relini [60,61]. (**b**) The proportion of taxa for which the only Ionian record in Relini [60,61] comes from the Strait of Messina biogeographic sector. Het., Heterobranchia. * indicates taxonomic groups that have already been included in previous publications [30,33].

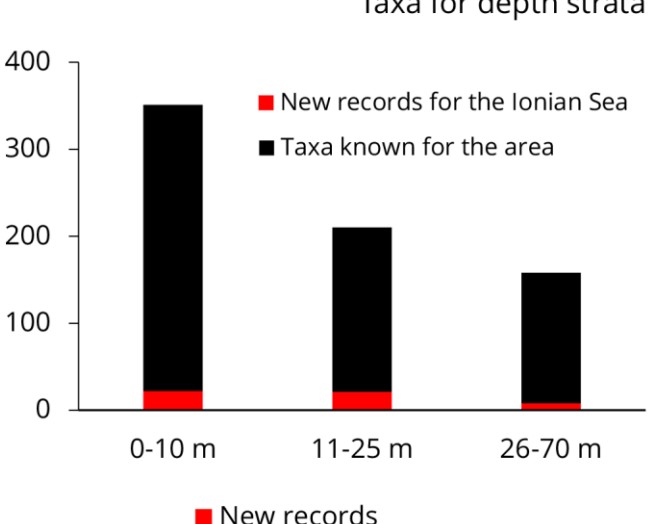

**Figure 7.** Total number of taxa and new records (in red) found by this study per depth strata.

Most specimens were collected by SCUBA diving (32%), free diving (31%) and by artisanal fishers' fishing gear (29%), while citizen science and recreational fishing gear, provided only 3% and 1% of the specimens, respectively. Regarding SCUBA and free diving, 79% of the specimens were collected physically either using the visual collection technique or by sampling substrate (biogenic substrate, pieces of rocks, and soft sediment), while the remaining 21% was only recorded photographically (mainly fish and organisms that could be identified only by the external morphology). Regarding sampling with fishing gears, most specimens were collected by set nylon trammel nets (63%), that were left 3–6 days in the water, and set monofilament gillnets (15%), that were left <24 h in the water (Table S2).

As for sampling substrate, most specimens we collected from rocky substrate (34%), coralligenous formations and rubble (22%), soft sediments including mud and sand (7%), while 9% were pelagic (Table S2).

Regarding the terrestrial near-shore vegetation, 81 species were found, 79 of which referred to the division Magnoliophyta and 2 to the division Polypodiophyta. Most species belonged to the order Asterales (16), Lamiales (13), Caryophyllales (10) and Poales (8). Regarding families, the most speciose ones were Asteraceae (13), Poaceae (8), Plantaginaceae (5), Apiaceae (4), and Crassulaceae (4). None of these species represent a new record for the area (Table S1).

### 3.3. Observations on Gelatinous Zooplankton

Gelatinous plankton blooms mostly occurred in spring and early summer, when complex communities developed. These communities were mainly composed of thaliaceans, scyphozoan and hydrozoan jellyfish, siphonophores, ctenophores, alciopid polychaetes and pterotracheid molluscs. Minor blooms involving fewer species also occurred throughout the year (Table 1 and Table S1).

Thaliaceans were very abundant in spring. *Salpa fusiformis* was the most abundant species and formed very dense blooms between March and April 2016; occasional isolated specimens were also found in autumn (November 2016). *Salpa maxima* bloomed later than *S. fusiformis* (April–May 2016). Isolate blastozooids of a rare helicosalpid (*Helicosalpa virgula*) were recorded on the 27 April 2016.

Regarding cnidarians, scyphozoan jellyfish mostly occurred in spring–summer. In 2016, *Aurelia solida* and *Chrysaora hysoscella* bloomed throughout April and May. Other scyphozoans were rarer, and only isolated specimens were found, and in particular *Rhizostoma pulmo* (February and June), and *Cotylorhiza tuberculata* (July–August). Regarding hydrozoan jellyfish, we found blooms of *Aequorea forskalea* and *Oceania armata* (April), and isolated specimens of *Geryonia proboscidalis* (April) and *Olindias muelleri* (September). Siphonophores were very common during the salp blooms in April and May, particularly *Halistemma rubrum*, *Nanomia bijuga*, *Forskalia formosa*, and *Hippopodius hippopus*.

As for ctenophores, the only blooming species was *Mnemiopsis leidyi* which occurred from November 2016 to January 2017. Other species were reported occasionally, including *Leucothea multicornis* (April–September), *Lampea pancerina* (April 2016, preying on *Salpa fusiformis*), and *Cestum veneris* (one specimen, May 2016).

**Table 1.** Complete list of marine species found during the project "Biodiversity MARE Tricase" with indication of depth or depth range (when multiple specimens were found) at which taxa were recorded during this study. For phenology, Rep. behav., reproductive behaviour; Rep. structures, reproductive structures. In the column New Record, * indicates a new record for the Ionian Sea, ‡ indicates that the only other record in the Ionian Sea is from the Messinian Strait, † indicates the easternmost record in the Mediterranean Sea, # indicates new species for science. For the records that were already reported in a separate publication, the reference is given at the end of the species name.

| Taxa | Depth | Phenology | New Record |
|:---:|:---:|:---:|:---:|
| **Kingdom BACTERIA** | | | |
| **Phylum CYANOBACTERIA** | | | |
| **Class CYANOPHYCEAE** | | | |
| **Order NOSTOCALES** | | | |
| **Family RIVULARIACEAE** | | | |
| *Rivularia* ind Roth ex Bornet & Flahault, 1886 | 0 m | - | |
| **Kingdom CHROMISTA** | | | |
| **Phylum FORAMINIFERA** | | | |
| **Class GLOBOTHALAMEA** | | | |
| **Order ROTALIIDA** | | | |
| **Family HOMOTREMATIDAE** | | | |
| *Miniacina miniacea* (Pallas, 1766) | 30 m | - | |
| **Order TEXTULARIIDA** | | | |
| **Family TEXTULARIIDAE** | | | |
| *Textularia* ind Defrance, 1824 | 25 m | - | |
| **Phylum RADIOZOA** | | | |
| **Class POLYCYSTINA** | | | |
| **Order COLLODARIA** | | | |
| *Collozoum* ind Haeckel, 1862 sensu Brandt, 1905 | 0–5 m | Occurrence (VI–VII) | |
| **Phylum OCHROPHYTA** | | | |
| **Class PHAEOPHYCEAE** | | | |
| **Order CUTLERIALES** | | | |
| **Family CUTLERIACEAE** | | | |
| *Cutleria multifida* (Turner) Greville | 1 m | - | |
| *Zanardinia typus* (Nardo) P.C.Silva | 5 m | - | |
| **Order DICTYOTALES** | | | |
| **Family DICTYOTACEAE** | | | |
| *Dictyopteris polypodioides* (A.P.De Candolle) J.V.Lamouroux | 1 m | - | |
| *Dictyota* cf. *implexa* (Desfontaines) J.V.Lamouroux | 20 m | - | |
| *Dictyota dichotoma* (Hudson) J.V.Lamouroux | 0–15 m | - | |
| *Padina pavonica* (Linnaeus) Thivy | 1–10 m | - | |
| **Order ECTOCARPALES** | | | |
| **Family SCYTOSIPHONACEAE** | | | |
| *Colpomenia sinuosa* (Mertens ex Roth) Derbès & Solier | 10 m | - | |
| *Scytosiphon lomentaria* (Lyngbye) Link | 1 m | - | |
| **Order FUCALES** | | | |
| **Family SARGASSACEAE** | | | |
| *Cystoseira compressa* (Esper) Gerloff & Nizamuddin | 1 m | - | |
| *Cystoseira foeniculacea* (Linnaeus) Greville | 20 m | - | |
| *Ericaria amentacea* (C.Agardh) Molinari & Guiry | 0–1 m | - | |
| *Sargassum vulgare* C.Agardh | 10 m | - | |
| **Order SPHACELARIALES** | | | |
| **Family CLADOSTEPHACEAE** | | | |

**Table 1.** *Cont.*

| Taxa | Depth | Phenology | New Record |
|---|---|---|---|
| *Cladostephus hirsutus* (Linnaeus) Boudouresque & M.Perret-Boudouresque ex Heesch & al. | 3 m | - | |
| **Family SPHACELARIACEAE** | | | |
| *Sphacelaria* sp. | 20 m | - | |
| **Family STYPOCAULACEAE** | | | |
| *Halopteris scoparia* (Linnaeus) Sauvageau | 10 m | - | |
| **Order SPOROCHNALES** | | | |
| **Family SPOROCHNACEAE** | | | |
| *Nereia filiformis* (J.Agardh) Zanardini | 40 m | - | |
| | | | |
| **Phylum CILIOPHORA** | | | |
| **Class HETEROTRICHEA** | | | |
| **Order HETEROTRICHIDA** | | | |
| **Family FOLLICULINIDAE** | | | |
| Folliculinidae ind (T.S. Wright, 1859) Dons, 1934 | 70 m | - | |
| **Class OLIGOHYMENOPHOREA** | | | |
| **Order SESSILIDA** | | | |
| **Family ZOOTHAMNIIDAE** | | | |
| *Zoothamnium niveum* Ehrenberg, 1838 | 10 m | - | * |
| | | | |
| **Kingdom PLANTAE** | | | |
| | | | |
| **Phylum RHODOPHYTA** | | | |
| **Class COMPSOPOGONOPHYCEAE** | | | |
| **Order ERYTHROPELTALES** | | | |
| **Family ERYTHROTRICHIACEAE** | | | |
| *Erythrotrichia carnea* (Dillwyn) J.Agardh | 0 m | - | |
| **Class FLORIDEOPHYCEAE** | | | |
| **Order BONNEMAISONIALES** | | | |
| **Family BONNEMAISONIACEAE** | | | |
| *Asparagopsis taxiformis* (Delile) Trevisan de Saint-Léon | 1 m | - | |
| **Order CERAMIALES** | | | |
| **Family CALLITHAMNIACEAE** | | | |
| *Callithamnion granulatum* (Ducluzeau) C.Agardh | 0 m | - | |
| *Crouania attenuata* (C.Agardh) J.Agardh | 10 m | - | |
| **Family CERAMIACEAE** | | | |
| *Ceramium* cf. *virgatum* Roth | 0 m | - | |
| *Ceramium diaphanum* (Lightfoot) Roth | 0 m | - | |
| *Gayliella mazoyerae* T.O.Cho, Fredericq & Hommersand | 2 m | - | |
| **Family DASYACEAE** | | | |
| *Dasya rigidula* (Kützing) Ardissone | 20 m | - | |
| **Family DELESSERIACEAE** | | | |
| *Hypoglossum hypoglossoides* (Stackhouse) Collins & Hervey | 1 m | - | |
| **Family RHODOMELACEAE** | | | |
| *Chondria* sp. (Montagne) | 0 m | - | |
| *Herposiphonia secunda* (C.Agardh) Ambronn | 20 m | - | |
| *Laurencia* cf. *microcladia* Kützing | 20 m | - | |
| *Laurencia* cf. *obtusa* (Hudson) J.V.Lamouroux | 1 m | - | |
| *Laurencia glandulifera* (Kützing) Kützing | 1 m | - | |
| *Osmundaria volubilis* (Linnaeus) R.E.Norris | 30 m | - | |
| *Osmundea truncata* (Kützing) K.W.Nam & Maggs | 0 m | - | |
| *Palisada perforata* (Bory) K.W.Nam | 2 m | - | |
| *Polysiphonia opaca* (C.Agardh) Moris & De Notaris | 0 m | - | |
| *Polysiphonia sertularioides* (Grateloup) J.Agardh | 0 m | - | |
| *Vertebrata fruticulosa* (Wulfen) Kuntze | 1 m | Rep. structures (IV) | |
| *Xiphosiphonia pennata* (C.Agardh) Savoie & G.W.Saunders | 0 m | - | |
| **Family WRANGELIACEAE** | | | |
| *Wrangelia penicillata* (C.Agardh) C.Agardh | 10 m | - | |

**Table 1.** *Cont.*

| Taxa | Depth | Phenology | New Record |
|---|---|---|---|
| **Order CORALLINALES** | | | |
| **Family CORALLINACEAE** | | | |
| *Ellisolandia elongata* (J.Ellis & Solander) K.R.Hind & G.W.Saunders | 0–1 m | - | |
| *Jania rubens* (Linnaeus) J.V.Lamouroux | 2 m | - | |
| *Jania virgata* (Zanardini) Montagne | 10–18 m | - | |
| **Family LITHOPHYLLACEAE** | | | |
| *Amphiroa rigida* J.V.Lamouroux | 2 m | - | |
| *Lithophyllum byssoides* (Lamarck) Foslie | 0 m | - | |
| *Lithophyllum* cf. *incrustans* Philippi | 3 m | - | |
| *Lithophyllum stictiforme* (J.E. Areschoug) Hauck | 5–10 m | - | |
| *Tenarea tortuosa* (Esper) Me.Lemoine | 1 m | - | |
| *Titanoderma trochanter* (Bory) Benhissoune, Boudouresque, Perret-Boudouresque & Verlaque | 1 m | - | |
| **Family MASTOPORACEAE** | | | |
| *Pneophyllum fragile* Kützing | 0 m | - | |
| **Order GELIDIALES** | | | |
| **Family GELIDIACEAE** | | | |
| *Gelidium spinosum* (S.G.Gmelin) P.C.Silva | 1 m | - | |
| **Family PTEROCLADIACEAE** | | | |
| *Pterocladiella capillacea* (S.G.Gmelin) Santelices & Hommersand | 0 m | - | |
| **Order GIGARTINALES** | | | |
| **Family CYSTOCLONIACEAE** | | | |
| *Hypnea musciformis* (Wulfen) J.V.Lamouroux | 2 m | - | |
| **Family GIGARTINACEAE** | | | |
| *Chondracanthus acicularis* (Roth) Fredericq | 0 m | - | |
| **Family PHYLLOPHORACEAE** | | | |
| *Phyllophora crispa* (Hudson) P.S.Dixon | 30 m | - | |
| *Schottera nicaeensis* (J.V.Lamouroux ex Duby) Guiry & Hollenberg | 1 m | - | |
| **Family RHIZOPHYLLIDACEAE** | | | |
| *Contarinia squamariae* (Meneghini) Denizot | 5 m | - | |
| **Family SPHAEROCOCCACEAE** | | | |
| *Sphaerococcus coronopifolius* Stackhouse | 7–10 m | - | |
| **Order HALYMENIALES** | | | |
| **Family HALYMENIACEAE** | | | |
| *Halymenia floresii* (Clemente) C.Agardh | 15 m | - | |
| **Order NEMALIALES** | | | |
| **Family GALAXAURACEAE** | | | |
| *Tricleocarpa fragilis* (Linnaeus) Huisman & R.A.Townsend | 10 m | - | |
| **Family LIAGORACEAE** | | | |
| *Liagora viscida* (Forsskål) C.Agardh | 0–2 m | - | |
| **Order NEMASTOMATALES** | | | |
| **Family NEMASTOMATACEAE** | | | |
| *Predaea ollivieri* Feldmann | 22 m | - | |
| **Family SCHIZYMENIACEAE** | | | |
| *Platoma cyclocolpum* (Montagne) F.Schmitz | 3 m | Rep. structures (VIII) | |
| **Order PEYSSONNELIALES** | | | |
| **Family PEYSSONNELIACEAE** | | | |
| *Peyssonnelia heteromorpha* (Zanardini) Athanasiadis | 10 m | - | |
| *Peyssonnelia rosa-marina* Boudouresque & Denizot | 20 m | - | |
| *Peyssonnelia rubra* (Greville) J.Agardh | 0 m | - | |
| *Peyssonnelia squamaria* (S.G.Gmelin) Decaisne ex J.Agardh | 5–20 m | - | |
| **Order RHODYMENIALES** | | | |
| **Family CHAMPIACEAE** | | | |
| *Gastroclonium clavatum* (Roth) Ardissone | 1 m | - | |
| **Family RHODYMENIACEAE** | | | |
| *Botryocladia* sp. | 30 m | - | |
| *Irvinea boergesenii* (Feldmann) R.J.Wilkes, L.M.McIvor & Guiry | 30 m | - | |

**Table 1.** *Cont.*

| Taxa | Depth | Phenology | New Record |
|---|---|---|---|
| **Phylum CHLOROPHYTA** | | | |
| **Class PYRAMIMONADOPHYCEAE** | | | |
| **Order PALMOPHYLLALES** | | | |
| **Family PALMOPHYLLACEAE** | | | |
| *Palmophyllum crassum* (Naccari) Rabenhorst | 4 m | - | |
| **Class ULVOPHYCEAE** | | | |
| **Order BRYOPSIDALES** | | | |
| **Family BRYOPSIDACEAE** | | | |
| *Bryopsis* cf. *pennata* J.V.Lamouroux | 1 m | - | |
| *Bryopsis cupressina* J.V.Lamouroux | 0 m | - | |
| **Family CAULERPACEAE** | | | |
| *Caulerpa cylindracea* Sonder | 8–45 m | - | |
| *Caulerpa prolifera* (Forsskål) J.V.Lamouroux | 15 m | - | |
| **Family CODIACEAE** | | | |
| *Codium bursa* (Olivi) C.Agardh | 8 m | - | |
| *Codium* cf. *vermilara* (Olivi) Delle Chiaje | 5–10 m | - | |
| *Codium coralloides* (Kützing) P.C.Silva | 8–10 m | - | |
| **Family HALIMEDACEAE** | | | |
| *Halimeda tuna* (J.Ellis & Solander) J.V.Lamouroux | 10–25 m | - | |
| **Family UDOTEACEAE** | | | |
| *Flabellia petiolata* (Turra) Nizamuddin | 10–15 m | - | |
| *Pseudochlorodesmis furcellata* (Zanardini) Børgesen | 3 m | - | |
| **Order CLADOPHORALES** | | | |
| **Family ANADYOMENACEAE** | | | |
| *Anadyomene stellata* (Wulfen) C.Agardh | 1–20 m | - | |
| **Family CLADOPHORACEAE** | | | |
| *Chaetomorpha linum* (O.F.Müller) Kützing | 0 m | - | |
| *Cladophora dalmatica* Kützing | 0 m | - | |
| *Cladophora laetevirens* (Dillwyn) Kützing | 0 m | - | |
| **Family VALONIACEAE** | | | |
| *Valonia utricularis* (Roth) C.Agardh | 2 m | - | |
| **Order DASYCLADALES** | | | |
| **Family POLYPHYSACEAE** | | | |
| *Acetabularia acetabulum* (Linnaeus) P.C.Silva | 3–10 m | - | |
| **Order ULVALES** | | | |
| **Family ULVACEAE** | | | |
| *Ulva linza* Linnaeus | 0 m | - | |
| | | | |
| **Phylum TRACHEOPHYTA** | | | |
| **Class MAGNOLIOPSIDA** | | | |
| **Order ALISMATALES** | | | |
| **Family CYMODOCEACEAE** | | | |
| *Cymodocea nodosa* (Ucria) Ascherson | 6 m | - | |
| **Family POSIDONIACEAE** | | | |
| *Posidonia oceanica* (Linnaeus) Delile | 10–22 m | - | |
| | | | |
| **Kingdom ANIMALIA** | | | |
| | | | |
| **Phylum CTENOPHORA** | | | |
| **Class TENTACULATA** | | | |
| **Order CESTIDA** | | | |
| **Family CESTIDAE** | | | |
| *Cestum veneris* Lesueur, 1813 | 3 m | Occurrence (V) | ‡ |
| **Order CYDIPPIDA** | | | |
| **Family LAMPEIDAE** | | | |
| *Lampea pancerina* (Chun, 1879) | 3 m | Occurrence (IV) | ‡† |
| **Order LOBATA** | | | |

**Table 1.** *Cont.*

| Taxa | Depth | Phenology | New Record |
|---|---|---|---|
| **Family BOLINOPSIDAE** | | | |
| *Mnemiopsis leidyi* A. Agassiz, 1865 | 1–5 m | Occurrence (XI–I) | |
| **Family LEUCOTHEIDAE** | | | |
| *Leucothea multicornis* (Quoy & Gaimard, 1824) | 0–3 m | Occurrence (IV–IX) | |
| | | | |
| **Phylum PORIFERA** | | | |
| **Class CALCAREA** | | | |
| **Order CLATHRINIDA** | | | |
| **Family CLATHRINIDAE** | | | |
| *Clathrina clathrus* (Schmidt, 1864) [33] | 40 m | - | |
| **Order LEUCOSOLENIDA** | | | |
| **Family SYCETTIDAE** | | | |
| *Sycon raphanus* Schmidt, 1862 [33] | 70 m | - | |
| **Class DEMOSPONGIAE** | | | |
| **Order AGELASIDA** | | | |
| **Family AGELASIDAE** | | | |
| *Agelas oroides* (Schmidt, 1864) [33] | 20–42 m | - | |
| **Family HYMERHABDIIDAE** | | | |
| *Hymerhabdia oxytrunca* Topsent, 1904 [33] | 20 m | - | * |
| *Prosuberites longispinus* Topsent, 1893 [33] | 20 m | - | |
| **Order AXINELLIDA** | | | |
| **Family AXINELLIDAE** | | | |
| *Axinella cannabina* (Esper, 1794) [33] | 20–50 m | - | |
| *Axinella damicornis* (Esper, 1794) [33] | 20–35 m | - | |
| *Axinella polypoides* Schmidt, 1862 [33] | 10 m | - | |
| *Axinella verrucosa* (Esper, 1794) [33] | 20–40 m | - | |
| **Family RASPAILIIDAE** | | | |
| *Didiscus stylifer* Tsurnamal, 1969 [33] | 20 m | - | |
| *Eurypon cinctum* Sarà, 1960 [33] | 20 m | - | * |
| *Eurypon clavatum* (Bowerbank, 1866) [33] | 20 m | - | |
| *Eurypon coronula* (Bowerbank, 1874) [33] | 20 m | - | * |
| *Eurypon gracile* Bertolino, Calcinai & Pansini, 2013 [33] | 20 m | - | |
| *Eurypon major* Sarà & Siribelli, 1960 [33] | 20 m | - | |
| *Eurypon obtusum* Vacelet, 1969 [33] | 20 m | - | * |
| *Eurypon viride* (Topsent, 1889) [33] | 20 m | - | |
| *Raspaciona aculeata* (Johnston, 1842) [33] | 20 m | - | |
| *Raspailia (Raspailia) viminalis* Schmidt, 1862 [33] | 70 m | - | * |
| **Family STELLIGERIDAE** | | | |
| *Halicnemia geniculata* Sarà, 1958 [33] | 20 m | - | * |
| **Order BIEMNIDA** | | | |
| **Family RHABDEREMIIDAE** | | | |
| *Rhabderemia gallica* van Soest & Hooper, 1993 [33] | 20 m | - | * |
| *Rhabderemia minutula* (Carter, 1876) [33] | 20 m | - | |
| **Order BUBARIDA** | | | |
| **Family BUBARIDAE** | | | |
| *Bubaris carcisis* Vacelet, 1969 [33] | 20 m | - | |
| *Bubaris vermiculata* (Bowerbank, 1866) [33] | 20 m | - | |
| *Monocrepidium vermiculatum* Topsent, 1898 [33] | 20 m | - | * |
| *Rhabdobaris implicata* Pulitzer-Finali, 1983 [33] | 20 m | - | * |
| **Family DESMANTHIDAE** | | | |
| *Desmanthus incrustans* (Topsent, 1889) [33] | 20 m | - | |
| **Family DICTYONELLIDAE** | | | |
| *Acanthella acuta* Schmidt, 1862 [33] | 25–30 m | - | |
| *Dictyonella incisa* (Schmidt, 1880) [33] | 10 m | - | |
| *Dictyonella obtusa* (Schmidt, 1862) [33] | 22 m | - | |
| **Order CHONDRILLIDA** | | | |
| **Family CHONDRILLIDAE** | | | |
| *Chondrilla nucula* Schmidt, 1862 [33] | 5–25 m | - | |

**Table 1.** *Cont.*

| Taxa | Depth | Phenology | New Record |
|---|---|---|---|
| **Order CHONDROSIIDA** | | | |
| **Family CHONDROSIIDAE** | | | |
| *Chondrosia reniformis* Nardo, 1847 [33] | 2–20 m | Spawning (VII) | |
| **Order CLIONAIDA** | | | |
| **Family CLIONAIDAE** | | | |
| *Cliona amplicavata* Rützler, 1974 [33] | 20 m | - | * |
| *Cliona burtoni* Topsent, 1932 [33] | 20 m | - | * |
| *Cliona celata* Grant, 1826 [33] | 5 m | - | |
| *Cliona schmidtii* (Ridley, 1881) [33] | 10 m | - | |
| *Cliona viridis* (Schmidt, 1862) [33] | 5–20 m | - | |
| *Cliothosa hancocki* (Topsent, 1888) [33] | 20 m | - | |
| *Spiroxya heteroclita* Topsent, 1896 [33] | 20 m | - | |
| *Spiroxya sarai* (Melone, 1965) [33] | 20 m | - | |
| **Family SPIRASTRELLIDAE** | | | |
| *Diplastrella bistellata* (Schmidt, 1862) [33] | 20 m | - | |
| *Diplastrella boeroi* Bertolino, Costa & Pansini, 2019 [33] | 20 m | - | # |
| *Spirastrella cunctatrix* Schmidt, 1868 [33] | 20 m | - | |
| *Spirastrella regina* Bertolino, Costa & Pansini, 2020 [33] | 20 m | - | # |
| **Order DICTYOCERATIDA** | | | |
| **Family DYSIDEIDAE** | | | |
| *Dysidea avara* (Schmidt, 1862) [33] | 20 m | - | |
| *Dysidea fragilis* (Montagu, 1814) [33] | 20 m | - | |
| **Family IRCINIIDAE** | | | |
| *Ircinia variabilis* (Schmidt, 1862) [33] | 10 m | - | |
| *Sarcotragus foetidus* Schmidt, 1862 [33] | 5–10 m | - | |
| *Sarcotragus spinosulus* Schmidt, 1862 [33] | 12 m | - | |
| **Family THORECTIDAE** | | | |
| *Fasciospongia cavernosa* (Schmidt, 1862) [33] | 5–20 m | - | |
| *Scalarispongia scalaris* (Schmidt, 1862) [33] | 10 m | - | |
| **Order HAPLOSCLERIDA** | | | |
| **Family CHALINIDAE** | | | |
| *Haliclona (Reniera) mediterranea* Griessinger, 1971 [33] | 5 m | - | |
| *Haliclona (Rhizoniera) sarai* (Pulitzer-Finali, 1969) [33] | 5 m | - | |
| *Haliclona (Soestella) mucosa* (Griessinger, 1971) [33] | 7 m | - | * |
| *Haliclona (Soestella) valliculata* (Griessinger, 1971) [33] | 5 m | - | * |
| **Family PETROSIIDAE** | | | |
| *Petrosia (Petrosia) clavata* (Esper, 1794) [33] | 20 m | - | |
| *Petrosia (Petrosia) ficiformis* (Poiret, 1789) [33] | 7–40 m | - | |
| *Petrosia (Strongylophora) pulitzeri* Pansini, 1996 [33] | 7 m | - | * |
| *Petrosia (Strongylophora) vansoesti* Boury-Esnault, Pansini & Uriz, 1994 [33] | 5 m | - | * |
| **Family PHLOEODICTYIDAE** | | | |
| *Oceanapia perforata* (Sarà, 1960) [33] | 7 m | - | * |
| **Order POECILOSCLERIDA** | | | |
| **Family COELOSPHAERIDAE** | | | |
| *Lissodendoryx (Anomodoryx) cavernosa* (Topsent, 1892) [33] | 20 m | - | |
| **Family CRAMBEIDAE** | | | |
| *Crambe crambe* (Schmidt, 1862) [33] | 5 m | - | |
| **Family CRELLIDAE** | | | |
| *Crella* sp. Gray, 1867 [33] | 20 m | - | |
| **Family ESPERIOPSIDAE** | | | |
| *Ulosa digitata* (Schmidt, 1866) [33] | 10 m | - | |
| **Family HYMEDESMIIDAE** | | | |
| *Hamigera hamigera* (Schmidt, 1862) [33] | 3 m | - | |
| *Hemimycale columella* (Bowerbank, 1874) [33] | 6 m | - | |
| *Phorbas dives* (Topsent, 1891) [33] | 20 m | - | |
| *Phorbas fictitius* (Bowerbank, 1866) [33] | 20 m | - | |
| *Phorbas tenacior* (Topsent, 1925) [33] | 20 m | - | |

**Table 1.** *Cont.*

| Taxa | Depth | Phenology | New Record |
|---|---|---|---|
| **Family MICROCIONIDAE** | | | |
| *Clathria (Clathria) coralloides* (Scopoli, 1772) [33] | 70 m | - | |
| *Clathria (Clathria) toxistricta* Topsent, 1925 [33] | 70 m | - | |
| *Clathria (Clathria) toxivaria* (Sarà, 1959) [33] | 20 m | - | |
| **Family MYCALIDAE** | | | |
| *Mycale (Mycale) lingua* (Bowerbank, 1866) [33] | 20 m | - | |
| *Mycale (Mycale) massa* (Schmidt, 1862) [33] | 20 m | - | |
| **Order SUBERITIDA** | | | |
| **Family HALICHONDRIIDAE** | | | |
| *Axinyssa aurantiaca* (Schmidt, 1864) [33] | 20 m | - | |
| *Halichondria (Halichondria)* cf. *panicea* (Pallas, 1766) [33] | 20 m | - | |
| *Halichondria (Halichondria) contorta* (Sarà, 1961) [33] | 10 m | - | * |
| *Spongosorites intricatus* (Topsent, 1892) [33] | 20 m | - | |
| **Family SUBERITIDAE** | | | |
| *Aaptos aaptos* (Schmidt, 1864) [33] | 4–10 m | - | |
| *Protosuberites epiphytum* (Lamarck, 1815) [33] | 20 m | - | |
| *Terpios gelatinosus* (Bowerbank, 1866) [33] | 20 m | - | |
| **Order TETHYIDA** | | | |
| **Family TETHYIDAE** | | | |
| *Tethya aurantium* (Pallas, 1766) [33] | 20 m | - | |
| *Tethya citrina* Sarà & Melone, 1965 [33] | 20 m | - | |
| **Family TIMEIDAE** | | | |
| *Timea stellata* (Bowerbank, 1866) [33] | 20 m | - | |
| *Timea unistellata* (Topsent, 1892) [33] | 20 m | - | |
| **Order TETRACTINELLIDA** | | | |
| **Family ANCORINIDAE** | | | |
| *Ancorina cerebrum* Schmidt, 1862 [33] | 20 m | - | * |
| *Dercitus (Stoeba) plicatus* (Schmidt, 1868) [33] | 20 m | - | |
| *Jaspis incrustans* (Topsent, 1890) [33] | 20 m | - | |
| *Jaspis johnstonii* (Schmidt, 1862) [33] | 20 m | - | |
| *Stelletta grubii* Schmidt, 1862 [33] | 20 m | - | |
| *Stelletta mediterranea* (Topsent, 1893) [33] | 20 m | - | * |
| *Stelletta stellata* Topsent, 1893 [33] | 20 m | - | |
| **Family CALTHROPELLIDAE** | | | |
| *Calthropella (Corticellopsis) stelligera* (Schmidt, 1868) [33] | - | - | * |
| **Family GEODIIDAE** | | | |
| *Erylus discophorus* (Schmidt, 1862) [33] | 20 m | - | |
| *Geodia cydonium* (Linnaeus, 1767) [33] | 20 m | - | |
| *Penares euastrum* (Schmidt, 1868) [33] | 20 m | - | |
| *Penares helleri* (Schmidt, 1864) [33] | 20 m | - | |
| **Family PACHASTRELLIDAE** | | | |
| *Pachastrella monilifera* Schmidt, 1868 [33] | 20 m | - | |
| *Triptolemma simplex* (Sarà, 1959) [33] | 20 m | - | |
| **Family TETILLIDAE** | | | |
| *Tetilla* sp. Schmidt, 1868 [33] | 20 m | - | |
| **Family THOOSIDAE** | | | |
| *Alectona millari* Carter, 1879 [33] | 20 m | - | |
| *Delectona madreporica* Bavestrello, Calcinai, Cerrano & Sarà, 1997 [33] | 20 m | - | * |
| **Family VULCANELLIDAE** | | | |
| *Poecillastra compressa* (Bowerbank, 1866) [33] | 20 m | - | |
| **Order VERONGIIDA** | | | |
| **Family APLYSINIDAE** | | | |
| *Aplysina aerophoba* (Nardo, 1833) [33] | 20 m | - | |
| *Aplysina cavernicola* (Vacelet, 1959) [33] | 70 m | - | |
| **Family IANTHELLIDAE** | | | |
| *Hexadella racovitzai* Topsent, 1896 [33] | 15 m | - | |
| **Class HOMOSCLEROMORPHA** | | | |
| **Order HOMOSCLEROPHORIDA** | | | |

**Table 1.** *Cont.*

| Taxa | Depth | Phenology | New Record |
|---|---|---|---|
| **Family OSCARELLIDAE** | | | |
| *Oscarella* cf. *tuberculata* (Schmidt, 1868) [33] | 5–15 m | - | * |
| **Family PLAKINIDAE** | | | |
| *Corticium candelabrum* Schmidt, 1862 [33] | 20 m | - | |
| *Plakina dilopha* Schulze, 1880 [33] | 20 m | - | * |
| *Plakina reducta* (Pulitzer-Finali, 1983) [33] | 20 m | - | |
| *Plakina trilopha* Schulze, 1880 [33] | 20 m | - | |
| *Plakortis simplex* Schulze, 1880 [33] | 20 m | - | |
| | | | |
| **Phylum CNIDARIA** | | | |
| **Class ANTHOZOA** | | | |
| **Order ACTINIARIA** | | | |
| **Family ACTINIIDAE** | | | |
| *Actinia mediterranea* Schmidt, 1971 | 0 m | - | |
| *Anemonia viridis* (Forsskål, 1775) | 10 m | - | |
| *Condylactis aurantiaca* (Delle Chiaje, 1825) | 28 m | - | |
| *Paranemonia cinerea* (Contarini, 1844) | 3 m | - | * |
| **Family AIPTASIIDAE** | | | |
| *Aiptasia mutabilis* (Gravenhorst, 1831) | 3 m | - | |
| *Exaiptasia diaphana* (Rapp, 1829) | 8 m | - | |
| **Family HALCAMPIDAE** | | | |
| *Halcampoides purpureus* (Studer, 1879) | 4 m | - | * |
| **Family HORMATHIIDAE** | | | |
| *Calliactis parasitica* (Couch, 1842) | 20–50 m | - | |
| **Family PHYMANTHIDAE** | | | |
| *Phymanthus pulcher* (Andrès, 1883) | 23 m | - | * |
| **Order ALCYONACEA** | | | |
| **Family ALCYONIIDAE** | | | |
| *Alcyonium* cf. *coralloides* (Pallas, 1766) | 45 m | - | ‡ |
| *Alcyonium palmatum* Pallas, 1766 | 60 m | - | |
| **Family CORALLIIDAE** | | | |
| *Corallium rubrum* (Linnaeus, 1758) | 60 m | - | |
| **Family CORNULARIIDAE** | | | |
| *Cornularia cornucopiae* (Pallas, 1766) | 5–70 m | - | * |
| **Family GORGONIIDAE** | | | |
| *Eunicella cavolini* (Koch, 1887) | 40 m | - | |
| *Leptogorgia sarmentosa* (Esper, 1789) | 30 m | - | |
| **Family PLEXAURIDAE** | | | |
| *Paramuricea clavata* (Risso, 1826) | 55 m | - | |
| **Order ANTIPATHARIA** | | | |
| **Family MYRIOPATHIDAE** | | | |
| *Antipathella subpinnata* (Ellis & Solander, 1786) | 80 m | - | |
| **Order PENICILLARIA** | | | |
| **Family ARACHNACTIDAE** | | | |
| *Arachnanthus* cf. *oligopodus* (Cerfontaine, 1891) | 4 m | - | |
| **Order PENNATULACEA** | | | |
| **Family PENNATULIDAE** | | | |
| *Pteroeides griseum* (Bohadsch, 1761) | 70 m | - | |
| **Family VIRGULARIIDAE** | | | |
| *Virgularia mirabilis* (Müller, 1776) | 70 m | - | * |
| **Order SCLERACTINIA** | | | |
| **Family CARYOPHYLLIIDAE** | | | |
| *Caryophyllia (Caryophyllia)* cf. *inornata* (Duncan, 1878) | 3 m | - | |
| *Paracyathus pulchellus* (Philippi, 1842) | 70 m | - | |
| *Phyllangia americana mouchezii* (Lacaze-Duthiers, 1897) | 40 m | - | |
| *Polycyathus muellerae* (Abel, 1959) | 4 m | - | |
| **Family DENDROPHYLLIIDAE** | | | |
| *Balanophyllia (Balanophyllia) europaea* (Risso, 1826) | 4–15 m | - | |

**Table 1.** *Cont.*

| Taxa | Depth | Phenology | New Record |
|---|---|---|---|
| **Family POCILLOPORIDAE** | | | |
| *Madracis pharensis* (Heller, 1868) | 15 m | - | |
| **Family SCLERACTINIA INCERTAE SEDIS** | | | |
| *Cladocora caespitosa* (Linnaeus, 1767) | 8 m | - | |
| **Order SPIRULARIA** | | | |
| **Family CERIANTHIDAE** | | | |
| *Cerianthus membranaceus* (Gmelin, 1791) | 10–20 m | - | |
| **Order ZOANTHARIA** | | | |
| **Family EPIZOANTHIDAE** | | | |
| *Epizoanthus* ind Gray, 1867 | 60 m | - | |
| **Family PARAZOANTHIDAE** | | | |
| *Parazoanthus axinellae* (Schmidt, 1862) | 21 m | - | |
| *Savalia savaglia* (Bertoloni, 1819) | 60 m | - | |
| **Class HYDROZOA** | | | |
| **Order ANTHOATHECATA** | | | |
| **Family CORYNIDAE** | | | |
| *Coryne pintneri* Schneider, 1897 | 10–60 m | Rep. structures (VII) | |
| **Family EUDENDRIIDAE** | | | |
| *Eudendrium merulum* Watson, 1985 | 25 m | - | |
| *Eudendrium moulouyensis* Marques, Peña Cantero & Vervoort, 2000 | 5 m | Rep. structures (IX) | |
| *Eudendrium racemosum* (Cavolini, 1785) | 20 m | - | |
| **Family OCEANIIDAE** | | | |
| *Corydendrium parasiticum* (Linnaeus, 1767) | 1 m | - | |
| *Oceania armata* Kölliker, 1853 | 1 m | Medusa, Eggs (IV) | ‡ |
| *Rhizogeton nudus* Broch, 1910 | 3 m | - | |
| *Turritopsis dohrnii* (Weismann, 1883) | 2–70 m | Medusa (VII-IX) | |
| **Family PANDEIDAE** | | | |
| *Amphinema rugosum* (Mayer, 1900) | 25 m | - | |
| **Family PENNARIIDAE** | | | |
| *Pennaria disticha* Goldfuss, 1820 | 3 m | - | |
| **Family PROBOSCIDACTYLIDAE** | | | |
| *Proboscidactyla ornata* (McCrady, 1859) | 70 m | Medusa (VI) | * |
| **Family TUBULARIIDAE** | | | |
| *Ectopleura wrighti* Petersen, 1979 | 20 m | - | |
| **Family ZANCLEIDAE** | | | |
| *Halocoryne epizoica* Hadzi, 1917 | 10–20 m | Medusa (VI) | |
| *Zanclea giancarloi* Boero, Bouillon & Gravili, 2000 | 70 m | - | |
| **Order LEPTOTHECATA** | | | |
| **Family AEQUOREIDAE** | | | |
| *Aequorea forskalea* Péron & Lesueur, 1810 | 0–5 m | Medusa (IV) | |
| **Family AGLAOPHENIIDAE** | | | |
| *Aglaophenia elongata* Meneghini, 1845 | 11–50 m | - | |
| *Aglaophenia kirchenpaueri* (Heller, 1868) | 50 m | Rep. structures (IV) | |
| *Aglaophenia octodonta* (Heller, 1868) | 1–5 m | Rep. structures (IV) | |
| *Aglaophenia tubiformis* Marktanner-Turneretscher, 1890 | 11 m | Rep. structures (V) | |
| **Family CAMPANULARIIDAE** | | | |
| *Clytia hemisphaerica* (Linnaeus, 1767) | 11 m | - | |
| *Clytia hummelincki* (Leloup, 1935) | 5 m | - | |
| *Clytia linearis* (Thorneley, 1900) | 30 m | - | |
| *Obelia dichotoma* (Linnaeus, 1758) | 3–4 m | - | |
| *Orthopyxis integra* (MacGillivray, 1842) | 0 m | - | |
| **Family HALECIIDAE** | | | |
| *Halecium petrosum* Stechow, 1919 | 5 m | - | |
| *Halecium pusillum* Sars, 1856 | 3 m | - | |
| **Family HALOPTERIDIDAE** | | | |
| *Antennella secundaria* (Gmelin, 1791) | 3–11 m | - | |
| *Antennella siliquosa* (Hincks, 1877) | 15 m | - | |
| **Family HEBELLIDAE** | | | |

**Table 1.** *Cont.*

| Taxa | Depth | Phenology | New Record |
|---|---|---|---|
| *Anthohebella parasitica* (Ciamician, 1880) | 1–2 m | - | |
| *Scandia gigas* (Pieper, 1884) | 10 m | - | |
| **Family KIRCHENPAUERIIDAE** | | | |
| *Kirchenpaueria pinnata* (Linnaeus, 1758) | 0–20 m | - | |
| **Family LAFOEIDAE** | | | |
| *Filellum serpens* (Hassall, 1848) | - | - | |
| **Family LAODICEIDAE** | | | |
| *Laodicea undulata* (Forbes & Goodsir, 1853) | 45 m | - | |
| **Family PLUMULARIIDAE** | | | |
| *Monotheca obliqua* (Johnston, 1847) | 2 m | - | |
| *Nemertesia ramosa* (Lamarck, 1816) | 70 m | - | |
| *Plumularia posidoniae* (Picard, 1952) | 12 m | - | |
| *Plumularia setacea* (Linnaeus, 1758) | 3–5 m | - | |
| **Family SERTULARELLIDAE** | | | |
| *Sertularella ellisii* (Deshayes & Milne Edwards, 1836) | 3–60 m | Rep. structures (VI) | |
| *Sertularella gayi* (Lamouroux, 1821) | 70 m | - | |
| *Sertularella mediterranea* Hartlaub, 1901 | 50 m | - | |
| **Family SERTULARIIDAE** | | | |
| *Dynamena disticha* (Bosc, 1802) | 2 m | - | |
| **Family SYNTHECIIDAE** | | | |
| *Synthecium evansi* (Ellis & Solander, 1786) | 70 m | - | |
| **Order LIMNOMEDUSAE** | | | |
| **Family GERYONIIDAE** | | | |
| *Geryonia proboscidalis* (Forsskål, 1775) | 3 m | Medusa (IV) | ‡ |
| **Family OLINDIIDAE** | | | |
| *Olindias muelleri* Haeckel, 1879 | 7 m | Medusa (IX) | |
| **Order SIPHONOPHORAE** | | | |
| **Family AGALMATIDAE** | | | |
| *Halistemma rubrum* (Vogt, 1852) | 0 m | Occurrence (IV) | * |
| *Nanomia bijuga* (Delle Chiaje, 1844) | 5 m | Occurrence (IV) | |
| **Family FORSKALIIDAE** | | | |
| *Forskalia formosa* Keferstein & Ehlers, 1860 | 1–2 m | Occurrence (IV) | ‡ |
| **Family HIPPOPODIIDAE** | | | |
| *Hippopodius hippopus* (Forsskål, 1776) | 1 m | Occurrence (IV-V) | |
| **Family PRAYIDAE** | | | |
| *Rosacea cymbiformis* (Delle Chiaje, 1830) | 0–5 m | Occurrence (IV-V) | |
| **Class SCYPHOZOA** | | | |
| **Order CORONATAE** | | | |
| **Family NAUSITHOIDAE** | | | |
| *Nausithoe* ind Kölliker, 1853 | 70 m | - | |
| **Order RHIZOSTOMEAE** | | | |
| **Family CEPHEIDAE** | | | |
| *Cotylorhiza tuberculata* (Macri, 1778) | 1 m | Medusa (VII) | |
| **Family RHIZOSTOMATIDAE** | | | |
| *Rhizostoma pulmo* (Macri, 1778) | 2 m | Medusa (II; VI) | |
| **Order SEMAEOSTOMEAE** | | | |
| **Family PELAGIIDAE** | | | |
| *Chrysaora hysoscella* (Linnaeus, 1767) | 0–2 m | Medusa (IV-V) | |
| **Family ULMARIDAE** | | | |
| *Aurelia solida* Browne, 1905 | 0–5 m | Medusa (IV-V) | |
| | | | |
| **Phylum HEMICHORDATA** | | | |
| **Class GRAPTOLITHOIDEA** | | | |
| **Order RHABDOPLEUROIDEA** | | | |
| **Family RHABDOPLEURIDAE** | | | |
| *Rhabdopleura recondita* Beli, Cameron and Piraino, 2018 | 70 m | - | |
| | | | |
| **Phylum ECHINODERMATA** | | | |

**Table 1.** *Cont.*

| Taxa | Depth | Phenology | New Record |
|---|---|---|---|
| **Class ASTEROIDEA** | | | |
| **Order FORCIPULATIDA** | | | |
| **Family ASTERIIDAE** | | | |
| *Coscinasterias tenuispina* (Lamarck, 1816) | 6 m | - | |
| *Marthasterias glacialis* (Linnaeus, 1758) | 3–15 m | - | |
| **Order PAXILLOSIDA** | | | |
| **Family ASTROPECTINIDAE** | | | |
| *Astropecten aranciacus* (Linnaeus, 1758) | 10–34 m | - | |
| **Order SPINULOSIDA** | | | |
| **Family ECHINASTERIDAE** | | | |
| *Echinaster (Echinaster) sepositus* (Retzius, 1783) | 60 m | - | |
| **Order VALVATIDA** | | | |
| **Family CHAETASTERIDAE** | | | |
| *Chaetaster longipes* (Bruzelius, 1805) | 70 m | - | |
| **Family GONIASTERIDAE** | | | |
| *Peltaster placenta* (Müller & Troschel, 1842) | 45 m | - | |
| **Family OPHIDIASTERIDAE** | | | |
| *Hacelia attenuata* Gray, 1840 | 25 m | - | |
| *Ophidiaster ophidianus* (Lamarck, 1816) | 10 m | - | |
| **Class CRINOIDEA** | | | |
| **Order COMATULIDA** | | | |
| **Family ANTEDONIDAE** | | | |
| *Antedon mediterranea* (Lamarck, 1816) | 45 m | - | |
| **Class ECHINOIDEA** | | | |
| **Order ARBACIOIDA** | | | |
| **Family ARBACIIDAE** | | | |
| *Arbacia lixula* (Linnaeus, 1758) | 1–12 m | - | |
| **Order CAMARODONTA** | | | |
| **Family ECHINIDAE** | | | |
| *Echinus melo* Lamarck, 1816 | 50 m | - | |
| **Family PARECHINIDAE** | | | |
| *Paracentrotus lividus* (Lamarck, 1816) | 1–10 m | - | |
| *Psammechinus microtuberculatus* (Blainville, 1825) | 34 m | - | |
| **Family TOXOPNEUSTIDAE** | | | |
| *Sphaerechinus granularis* (Lamarck, 1816) | 8 m | - | |
| **Order CIDAROIDA** | | | |
| **Family CIDARIDAE** | | | |
| *Cidaris cidaris* (Linnaeus, 1758) | 70 m | - | |
| *Stylocidaris affinis* (Philippi, 1845) | 70 m | - | |
| **Order CLYPEASTEROIDA** | | | |
| **Family FIBULARIIDAE** | | | |
| *Echinocyamus pusillus* (O.F. Müller, 1776) | 18 m | - | |
| **Order DIADEMATOIDA** | | | |
| **Family DIADEMATIDAE** | | | |
| *Centrostephanus longispinus* (Philippi, 1845) | 50 m | - | |
| **Class HOLOTHUROIDEA** | | | |
| **Order HOLOTHURIIDA** | | | |
| **Family HOLOTHURIIDAE** | | | |
| *Holothuria (Holothuria) tubulosa* Gmelin, 1791 | 7 m | - | |
| *Holothuria (Panningothuria) forskali* Delle Chiaje, 1823 | 2 m | - | |
| *Holothuria (Platyperona) sanctori* Delle Chiaje, 1823 | 4 m | - | ‡ |
| **Order SYNALLACTIDA** | | | |
| **Family STICHOPODIDAE** | | | |
| *Parastichopus regalis* (Cuvier, 1817) | 60 m | - | |
| **Class OPHIUROIDEA** | | | |
| **Order AMPHILEPIDIDA** | | | |
| **Family OPHIOTRICHIDAE** | | | |
| *Ophiothrix fragilis* (Abildgaard in O.F. Müller, 1789) | 30–50 m | - | |

**Table 1.** *Cont.*

| Taxa | Depth | Phenology | New Record |
|---|---|---|---|
| **Order OPHIACANTHIDA** | | | |
| **Family OPHIODERMATIDAE** | | | |
| *Ophioderma longicaudum* (Bruzelius, 1805) | 10 m | - | |
| | | | |
| **Phylum CHORDATA** | | | |
| **Class ACTINOPTERYGII** | | | |
| **Order ANGUILLIFORMES** | | | |
| **Family CONGRIDAE** | | | |
| *Conger conger* (Linnaeus, 1758) | 2–4 m | - | |
| **Family MURAENIDAE** | | | |
| *Muraena helena* Linnaeus, 1758 | 8–11 m | - | |
| **Order ATHERINIFORMES** | | | |
| **Family ATHERINIDAE** | | | |
| *Atherina boyeri* Risso, 1810 | 2 m | - | |
| *Atherina hepsetus* Linnaeus, 1758 | 2 m | - | |
| **Order AULOPIFORMES** | | | |
| **Family SYNODONTIDAE** | | | |
| *Synodus saurus* (Linnaeus, 1758) | 1–10 m | - | |
| **Order BELONIFORMES** | | | |
| **Family BELONIDAE** | | | |
| *Belone belone* (Linnaeus, 1760) | 1 m | - | |
| **Order CLUPEIFORMES** | | | |
| **Family CLUPEIDAE** | | | |
| *Sardina pilchardus* (Walbaum, 1792) | 25 m | - | |
| *Sardinella aurita* Valenciennes, 1847 | - | - | |
| **Family ENGRAULIDAE** | | | |
| *Engraulis encrasicolus* (Linnaeus, 1758) | 1–5 m | - | |
| **Order GADIFORMES** | | | |
| **Family MERLUCCIIDAE** | | | |
| *Merluccius merluccius* (Linnaeus, 1758) | - | - | |
| **Family PHYCIDAE** | | | |
| *Phycis phycis* (Linnaeus, 1766) | 30 m | - | |
| **Order LOPHIIFORMES** | | | |
| **Family LOPHIIDAE** | | | |
| *Lophius* ind Linnaeus, 1758 | 60 m | - | |
| **Order MUGILIFORMES** | | | |
| **Family MUGILIDAE** | | | |
| *Chelon auratus* (Risso, 1810) | 2 m | - | |
| *Chelon labrosus* (Risso, 1827) | 1 m | - | |
| **Order PERCIFORMES** | | | |
| **Family APOGONIDAE** | | | |
| *Apogon imberbis* (Linnaeus, 1758) | 5 m | - | |
| **Family BLENNIIDAE** | | | |
| *Aidablennius sphynx* (Valenciennes, 1836) | 1 m | - | |
| *Coryphoblennius galerita* (Linnaeus, 1758) | 0 m | - | |
| *Lipophrys trigloides* (Valenciennes, 1836) | 1 m | - | |
| *Microlipophrys canevae* (Vinciguerra, 1880) | 1 m | - | |
| *Parablennius gattorugine* (Linnaeus, 1758) | 8–11 m | - | |
| *Parablennius incognitus* (Bath, 1968) | 2 m | - | |
| *Parablennius rouxi* (Cocco, 1833) | 10 m | - | |
| *Parablennius sanguinolentus* (Pallas, 1814) | 0–3 | - | |
| *Parablennius zvonimiri* (Kolombatovic, 1892) | 3 m | - | |
| **Family CARANGIDAE** | | | |
| *Lichia amia* (Linnaeus, 1758) | 1 m | Sperm (XI) | |
| *Seriola dumerili* (Risso, 1810) | 2 m | - | |
| *Trachinotus ovatus* (Linnaeus, 1758) | 3 m | - | |
| *Trachurus trachurus* (Linnaeus, 1758) | 60 m | - | |
| **Family CENTRACANTHIDAE** | | | |

**Table 1.** *Cont.*

| Taxa | Depth | Phenology | New Record |
|---|---|---|---|
| *Spicara maena* (Linnaeus, 1758) | - | - | |
| *Spicara smaris* (Linnaeus, 1758) | 7–25 m | Rep. behav. (IV) | |
| **Family GOBIIDAE** | | | |
| *Gobius cobitis* Pallas, 1814 | 0–3 | - | |
| *Gobius geniporus* Valenciennes, 1837 | 7 m | - | |
| *Gobius incognitus* Kovačić & Sanda, 2016 | 1–11 m | - | |
| **Family LABRIDAE** | | | |
| *Coris julis* (Linnaeus, 1758) | 1–15 m | - | |
| *Labrus merula* Linnaeus, 1758 | 20 m | - | |
| *Symphodus doderleini* Jordan, 1890 | 3 m | - | |
| *Symphodus mediterraneus* (Linnaeus, 1758) | 25 m | - | |
| *Symphodus ocellatus* (Linnaeus, 1758) | 2 m | - | |
| *Symphodus roissali* (Risso, 1810) | 2 m | - | |
| *Symphodus rostratus* (Bloch, 1791) | 5 m | - | |
| *Symphodus tinca* (Linnaeus, 1758) | 3 m | Rep. behav. (VI-VII) | |
| *Thalassoma pavo* (Linnaeus, 1758) | 1–5 m | - | |
| *Xyrichtys novacula* (Linnaeus, 1758) | 18 m | - | |
| **Family MULLIDAE** | | | |
| *Mullus barbatus barbatus* Linnaeus, 1758 | 60 m | - | |
| *Mullus surmuletus* Linnaeus, 1758 | 60 m | - | |
| **Family POMACENTRIDAE** | | | |
| *Chromis chromis* (Linnaeus, 1758) | 1–20 m | Rep. behav. (VIII) | |
| **Family POMATOMIDAE** | | | |
| *Pomatomus saltatrix* (Linnaeus, 1766) | - | - | |
| **Family SCARIDAE** | | | |
| *Sparisoma cretense* (Linnaeus, 1758) | 3 m | - | |
| **Family SCIAENIDAE** | | | |
| *Sciaena umbra* Linnaeus, 1758 | 23 m | - | |
| **Family SCOMBRIDAE** | | | |
| *Auxis rochei rochei* (Risso, 1810) | - | - | |
| *Euthynnus alletteratus* (Rafinesque, 1810) | 1 m | - | |
| *Sarda sarda* (Bloch, 1793) | 20 m | - | |
| *Scomber colias* Gmelin, 1789 | 18 m | - | |
| *Scomber scombrus* Linnaeus, 1758 | 60 m | - | |
| **Family SERRANIDAE** | | | |
| *Anthias anthias* (Linnaeus, 1758) | 40 m | - | |
| *Epinephelus costae* (Steindachner, 1878) | 30 m | - | |
| *Epinephelus marginatus* (Lowe, 1834) | 4 m | - | |
| *Serranus cabrilla* (Linnaeus, 1758) | 4–15 m | - | |
| *Serranus hepatus* (Linnaeus, 1758) | 7–25 m | - | |
| *Serranus scriba* (Linnaeus, 1758) | 2–18 m | - | |
| **Family SPARIDAE** | | | |
| *Boops boops* (Linnaeus, 1758) | 1–10 m | - | |
| *Dentex dentex* (Linnaeus, 1758) | - | - | |
| *Diplodus puntazzo* (Walbaum, 1792) | 5–15 m | - | |
| *Diplodus sargus* (Linnaeus, 1758) | 7 m | - | |
| *Diplodus vulgaris* (Geoffroy Saint-Hilaire, 1817) | 5–15 m | - | |
| *Lithognathus mormyrus* (Linnaeus, 1758) | 7 m | - | |
| *Oblada melanura* (Linnaeus, 1758) | 1 m | - | |
| *Pagellus acarne* (Risso, 1827) | 20 m | - | |
| *Pagellus bogaraveo* (Brünnich, 1768) | 0–3 | - | |
| *Pagellus erythrinus* (Linnaeus, 1758) | 20 m | - | |
| *Pagrus pagrus* (Linnaeus, 1758) | 0–3 | - | |
| *Sarpa salpa* (Linnaeus, 1758) | 30 m | - | |
| *Sparus aurata* Linnaeus, 1758 | 2 m | - | |
| *Spondyliosoma cantharus* (Linnaeus, 1758) | 0–2 m | - | |
| **Family SPHYRAENIDAE** | | | |
| *Sphyraena sphyraena* (Linnaeus, 1758) | 30 m | - | |

**Table 1.** *Cont.*

| Taxa | Depth | Phenology | New Record |
|---|---|---|---|
| **Family TRACHINIDAE** | | | |
| *Trachinus araneus* Cuvier, 1829 | 25 m | - | |
| **Family TRIPTERYGIIDAE** | | | |
| *Tripterygion tripteronotum* (Risso, 1810) | 0–3 | - | |
| **Family URANOSCOPIDAE** | | | |
| *Uranoscopus scaber* Linnaeus, 1758 | 30 m | - | |
| **Order PLEURONECTIFORMES** | | | |
| **Family BOTHIDAE** | | | |
| *Bothus podas* (Delaroche, 1809) | 1–4 m | - | |
| **Family SOLEIDAE** | | | |
| *Microchirus ocellatus* (Linnaeus, 1758) | 20 m | - | |
| *Solea solea* (Linnaeus, 1758) | 20 m | - | |
| **Order SCORPAENIFORMES** | | | |
| **Family DACTYLOPTERIDAE** | | | |
| *Dactylopterus volitans* (Linnaeus, 1758) | - | - | |
| **Family SCORPAENIDAE** | | | |
| *Scorpaena maderensis* Valenciennes, 1833 | 3–15 m | - | |
| *Scorpaena notata* Rafinesque, 1810 | 20 m | - | |
| *Scorpaena porcus* Linnaeus, 1758 | 16 m | - | |
| *Scorpaena scrofa* Linnaeus, 1758 | 25 m | - | |
| **Family TRIGLIDAE** | | | |
| *Chelidonichthys lastoviza* (Bonnaterre, 1788) | - | - | |
| **Order TETRAODONTIFORMES** | | | |
| **Family BALISTIDAE** | | | |
| *Balistes capriscus* Gmelin, 1789 | 15 m | - | |
| **Order ZEIFORMES** | | | |
| **Family ZEIDAE** | | | |
| *Zeus faber* Linnaeus, 1758 | 67 m | - | |
| **Class ASCIDIACEA** | | | |
| **Order APLOUSOBRANCHIA** | | | |
| **Family CLAVELINIDAE** | | | |
| *Pycnoclavella* ind Garstang, 1891 | 15 m | - | * |
| **Family DIDEMNIDAE** | | | |
| *Diplosoma spongiforme* (Giard, 1872) | 1–60 m | - | |
| **Family POLYCITORIDAE** | | | |
| *Polycitor crystallinus* (Renier, 1804) | 30 m | - | |
| **Family POLYCLINIDAE** | | | |
| *Aplidium* ind Savigny, 1816 | 15 m | - | |
| **Order PHLEBOBRANCHIA** | | | |
| **Family ASCIDIIDAE** | | | |
| *Ascidia mentula* Müller, 1776 | 4 m | - | |
| **Order STOLIDOBRANCHIA** | | | |
| **Family PYURIDAE** | | | |
| *Halocynthia papillosa* (Linnaeus, 1767) | 5–30 m | - | |
| *Microcosmus* ind Heller, 1877 | 20 m | - | |
| *Pyura tessellata* (Forbes, 1848) | 60 m | - | |
| **Family STYELIDAE** | | | |
| *Botrylloides niger* Herdman, 1886 | 1–3 m | - | |
| **Class ELASMOBRANCHII** | | | |
| **Order MYLIOBATIFORMES** | | | |
| **Family DASYATIDAE** | | | |
| *Dasyatis pastinaca* (Linnaeus, 1758) | 20 m | - | |
| **Order RAJIFORMES** | | | |
| **Family RAJIDAE** | | | |
| *Raja brachyura* Lafont, 1871 | 20 m | - | |
| *Raja miraletus* Linnaeus, 1758 | 67 m | Eggs (V) | |
| **Order TORPEDINIFORMES** | | | |
| **Family TORPEDINIDAE** | | | |

**Table 1.** *Cont.*

| Taxa | Depth | Phenology | New Record |
|---|---|---|---|
| *Torpedo marmorata* Risso, 1810 | 5 m | - | |
| **Class LEPTOCARDII** | | | |
| **Family BRANCHIOSTOMATIDAE** | | | |
| *Branchiostoma lanceolatum* (Pallas, 1774) | 7–11 m | - | |
| **Class MAMMALIA** | | | |
| **Order CETARTIODACTYLA** | | | |
| **Family DELPHINIDAE** | | | |
| *Tursiops truncatus* (Montagu, 1821) | 0 m | - | |
| **Class REPTILIA** | | | |
| **Order TESTUDINES** | | | |
| **Family CHELONIIDAE** | | | |
| *Caretta caretta* (Linnaeus, 1758) | 2 m | - | |
| **Class THALIACEA** | | | |
| **Order SALPIDA** | | | |
| **Family SALPIDAE** | | | |
| *Helicosalpa virgula* (Vogt, 1854)† | 3 m | Occurrence (IV) | *† |
| *Salpa fusiformis* Cuvier, 1804 | 0–5 m | Occurrence (III–IV) | |
| *Salpa maxima* Forskål, 1775 | 0–5 m | Occurrence (IV–V) | |
| | | | |
| **Phylum NEMATODA** | | | |
| **Class CHROMADOREA** | | | |
| **Order RHABDITIDA** | | | |
| **Family ANISAKIDAE** | | | |
| *Anisakis* ind Dujardin, 1845 | - | - | |
| | | | |
| **Phylum ARTHROPODA** | | | |
| **Class COLLEMBOLA** | | | |
| **Family NEANURIDAE** | | | |
| *Anurida maritima* (Guérin-Méneville, 1836) | 0 m | - | |
| **Class HEXANAUPLIA** | | | |
| **Family SACCULINIDAE** | | | |
| *Sacculina* ind Thompson, 1836 | 1 m | - | |
| **Order LEPADIFORMES** | | | |
| **Family LEPADIDAE** | | | |
| *Lepas (Anatifa) pectinata* Spengler, 1793 | 0 m | - | |
| **Order SIPHONOSTOMATOIDA** | | | |
| **Family PENNELLIDAE** | | | |
| *Pennella* ind Oken, 1815 | - | - | |
| **Class MALACOSTRACA** | | | |
| **Order AMPHIPODA** | | | |
| **Family CAPRELLIDAE** | | | |
| *Caprella* ind Lamarck, 1801 | 3–5 m | - | |
| *Phtisica marina* Slabber, 1769 | 5 m | - | |
| *Pseudoprotella phasma* (Montagu, 1804) | 5 m | - | |
| **Family ISCHYROCERIDAE** | | | |
| *Jassa marmorata* Holmes, 1905 | 2 m | - | |
| **Family PHRONIMIDAE** | | | |
| *Phronima sedentaria* (Forskål, 1775) | 3 m | - | |
| **Order DECAPODA** | | | |
| **Family ALPHEIDAE** | | | |
| *Alpheus dentipes* Guérin, 1832 | - | Eggs (VI) | |
| *Synalpheus gambarelloides* (Nardo, 1847) | - | - | |
| **Family CALAPPIDAE** | | | |
| *Calappa granulata* (Linnaeus, 1758) | 40–60 m | Eggs (II) | |
| **Family CARCINIDAE** | | | |
| *Xaiva biguttata* (Risso, 1816) | 7 m | - | |
| **Family DIOGENIDAE** | | | |
| *Calcinus tubularis* (Linnaeus, 1767) | 1 m | - | |

**Table 1.** *Cont.*

| Taxa | Depth | Phenology | New Record |
|---|---|---|---|
| *Clibanarius erythropus* (Latreille, 1818) | 1 m | - | |
| *Dardanus arrosor* (Herbst, 1796) | 70 m | - | |
| *Dardanus calidus* (Risso, 1827 in [Risso, 1826–1827]) | 20–50 m | Eggs (VIII) | |
| *Diogenes pugilator* (P. Roux, 1829) complex | 7 m | - | |
| *Paguristes eremita* (Linnaeus, 1767) | - | - | |
| **Family DORIPPIDAE** | | | |
| *Medorippe lanata* (Linnaeus, 1767) | 60 m | - | |
| **Family DROMIIDAE** | | | |
| *Dromia personata* (Linnaeus, 1758) | 5–50 m | Eggs (VIII) | |
| **Family EPIALTIDAE** | | | |
| *Acanthonyx lunulatus* (Risso, 1816) | 1 m | - | |
| *Herbstia condyliata* (Fabricius, 1787) | 4 m | - | |
| *Pisa* cf. *armata* (Latreille, 1803) | 70 m | - | |
| *Pisa* cf. *nodipes* Leach, 1815 | 60 m | - | |
| **Family ERIPHIIDAE** | | | |
| *Eriphia verrucosa* (Forskål, 1775) | 2 m | Eggs (VIII) | |
| **Family GALATHEIDAE** | | | |
| *Galathea squamifera* Leach, 1814 [in Leach, 1813–1815] | 5 m | - | |
| **Family GONEPLACIDAE** | | | |
| *Goneplax rhomboides* (Linnaeus, 1758) | 60 m | - | |
| **Family GRAPSIDAE** | | | |
| *Pachygrapsus marmoratus* (Fabricius, 1787) | 0 m | Eggs (VI) | |
| **Family HOMOLIDAE** | | | |
| *Homola barbata* (Fabricius, 1793) | 60 m | Eggs (VI) | |
| **Family INACHIDAE** | | | |
| *Macropodia* ind Leach, 1814 [in Leach, 1813–1815] | 20 m | - | |
| **Family LEUCOSIIDAE** | | | |
| *Ebalia edwardsii* O.G. Costa, 1838 [in O.G. Costa & A. Costa, 1838–1871] | 25 m | - | |
| *Ilia nucleus* (Linnaeus, 1758) | 7 m | - | |
| **Family MAJIDAE** | | | |
| *Maja crispata* Risso, 1827 in [Risso, 1826–1827] | - | - | |
| *Maja squinado* (Herbst, 1788) | 50 m | Eggs (VI;IX) | |
| *Neomaja goltziana* (d'Oliveira, 1889) | - | - | |
| **Family PAGURIDAE** | | | |
| *Cestopagurus timidus* (P. Roux, 1830 [in P. Roux, 1828–1830]) | 5 m | - | |
| *Pagurus anachoretus* Risso, 1827 in [Risso, 1826–1827] | - | - | |
| **Family PALAEMONIDAE** | | | |
| *Brachycarpus biunguiculatus* (H. Lucas, 1846) | 5 m | - | |
| *Palaemon* cf. *serratus* (Pennant, 1777) | 1 m | Eggs (VI) | |
| *Palaemon elegans* Rathke, 1836 | 0–1 m | - | |
| *Pontonia pinnophylax* (Otto, 1821) | 10 m | - | |
| **Family PALINURIDAE** | | | |
| *Palinurus elephas* (Fabricius, 1787) | 70 m | - | |
| **Family PARTHENOPIDAE** | | | |
| *Spinolambrus macrochelos* (Herbst, 1790 [in Herbst, 1782–1790]) | 70 m | - | |
| **Family PENAEIDAE** | | | |
| *Penaeus kerathurus* (Forskål, 1775) | 20 m | - | |
| **Family PERCNIDAE** | | | |
| *Percnon gibbesi* (H. Milne Edwards, 1853) | 1 m | - | |
| **Family PILUMNIDAE** | | | |
| *Pilumnus hirtellus* (Linnaeus, 1761) | 50 m | - | |
| *Pilumnus spinifer* H. Milne Edwards, 1834 | 50 m | Eggs (V) | |
| *Pilumnus villosissimus* (Rafinesque, 1814) | 0 m | - | |
| **Family POLYBIIDAE** | | | |
| *Liocarcinus vernalis* (Risso, 1827 in [Risso, 1826–1827]) | 0 m | - | |
| **Family PORTUNIDAE** | | | |
| *Callinectes sapidus* Rathbun, 1896 | - | - | |

**Table 1.** *Cont.*

| Taxa | Depth | Phenology | New Record |
|---|---|---|---|
| *Portunus hastatus* (Linnaeus, 1767) | 2–30 m | - | |
| **Family PROCESSIDAE** | | | |
| *Processa* ind Leach, 1815 [in Leach, 1815–1875] | 3 m | - | |
| **Family SCYLLARIDAE** | | | |
| *Scyllarides latus* (Latreille, 1803) | 60 m | Eggs (VI) | |
| **Family STENOPODIDAE** | | | |
| *Stenopus spinosus* Risso, 1827 in [Risso, 1826–1827] | 3 m | - | |
| **Family XANTHIDAE** | | | |
| *Paractaea monodi* Guinot, 1969 | 5 m | - | |
| *Xantho granulicarpus* Forest in Drach & Forest, 1953 | 0 m | - | |
| *Xantho poressa* (Olivi, 1792) | 0 m | - | |
| **Order ISOPODA** | | | |
| **Family GNATHIIDAE** | | | |
| *Gnathia* ind Leach, 1814 | - | - | |
| **Family IDOTEIDAE** | | | |
| *Idotea metallica* Bosc, 1802 | 0 m | - | |
| **Family LIGIIDAE** | | | |
| *Ligia italica* Fabricius, 1798 | 0 m | - | |
| **Order STOMATOPODA** | | | |
| **Family SQUILLIDAE** | | | |
| *Squilla mantis* (Linnaeus, 1758) | 60 m | - | |
| | | | |
| **Phylum PLATYHELMINTHES** | | | |
| **Order POLYCLADIDA** | | | |
| **Family EURYLEPTIDAE** | | | |
| *Prostheceraeus giesbrechtii* Lang, 1884 | 4–16 m | - | ‡ |
| **Family PLANOCERIDAE** | | | |
| *Planocera ceratommata* (Palombi, 1936) | 8 m | - | * |
| **Family PSEUDOCEROTIDAE** | | | |
| *Pseudoceros maximus-type A* Lang, 1884 | 15 m | - | * |
| *Pseudoceros velutinus* (Blanchard, 1847) | 2 m | - | * |
| *Thysanozoon brocchii* (Risso, 1818) | 8 m | - | * |
| **Class MONOGENEA** | | | |
| **Order MAZOCRAEIDEA** | | | |
| **Family HEXOSTOMATIDAE** | | | |
| *Hexostoma thynni* (Delaroche, 1811) Rafinesque, 1815 | - | - | |
| | | | |
| **Phylum MOLLUSCA** | | | |
| **Class BIVALVIA** | | | |
| **Order ADAPEDONTA** | | | |
| **Family HIATELLIDAE** | | | |
| *Hiatella arctica* (Linnaeus, 1767) | 1 m | - | |
| **Order ARCIDA** | | | |
| **Family ARCIDAE** | | | |
| *Arca noae* Linnaeus, 1758 | 20 m | - | |
| *Barbatia barbata* (Linnaeus, 1758) | - | - | |
| **Family GLYCYMERIDIDAE** | | | |
| *Glycymeris bimaculata* (Poli, 1795) | 18 m | - | |
| **Family NOETIIDAE** | | | |
| *Striarca lactea* (Linnaeus, 1758) | 50 m | - | |
| **Order CARDIIDA** | | | |
| **Family CARDIIDAE** | | | |
| *Laevicardium crassum* (Gmelin, 1791) | 18 m | - | |
| *Papillicardium papillosum* (Poli, 1791) | 10 m | - | |
| **Family TELLINIDAE** | | | |
| *Moerella donacina* (Linnaeus, 1758) | 18 m | - | |
| *Peronaea planata* (Linnaeus, 1758) | 18 m | - | |

**Table 1.** *Cont.*

| Taxa | Depth | Phenology | New Record |
|---|---|---|---|
| **Order CARDITIDA** | | | |
| **Family CARDITIDAE** | | | |
| *Cardita calyculata* (Linnaeus, 1758) | - | - | |
| **Order GALEOMMATIDA** | | | |
| **Family LASAEIDAE** | | | |
| *Lasaea rubra* (Montagu, 1803) | - | - | |
| **Order GASTROCHAENIDA** | | | |
| **Family GASTROCHAENIDAE** | | | |
| *Rocellaria dubia* (Pennant, 1777) | 2–5 m | - | |
| **Order LIMIDA** | | | |
| **Family LIMIDAE** | | | |
| *Lima lima* (Linnaeus, 1758) | 4 m | - | |
| *Limaria tuberculata* (Olivi, 1792) | 5 m | - | |
| **Order MYTILIDA** | | | |
| **Family MYTILIDAE** | | | |
| *Lithophaga lithophaga* (Linnaeus, 1758) | 2 m | - | |
| *Musculus costulatus* (Risso, 1826) | - | - | |
| *Mytilus galloprovincialis* Lamarck, 1819 | 1 m | - | |
| **Order OSTREIDA** | | | |
| **Family GRYPHAEIDAE** | | | |
| *Neopycnodonte cochlear* (Poli, 1795) | 4 m | - | |
| **Family PINNIDAE** | | | |
| *Pinna nobilis* Linnaeus, 1758 | 3–20 m | - | |
| **Family PTERIIDAE** | | | |
| *Pteria hirundo* (Linnaeus, 1758) | 50 m | - | |
| **Order PECTINIDA** | | | |
| **Family ANOMIIDAE** | | | |
| *Anomia ephippium* Linnaeus, 1758 | 20 m | - | |
| **Family PECTINIDAE** | | | |
| *Aequipecten opercularis* (Linnaeus, 1758) | 20 m | - | |
| *Manupecten pesfelis* (Linnaeus, 1758) | - | - | |
| *Mimachlamys varia* (Linnaeus, 1758) | 10 m | - | |
| *Palliolum incomparabile* (Risso, 1826) | 50 m | - | |
| *Pecten jacobaeus* (Linnaeus, 1758) | - | - | |
| *Talochlamys multistriata* (Poli, 1795) | - | - | |
| **Family SPONDYLIDAE** | | | |
| *Spondylus gaederopus* Linnaeus, 1758 | 30 m | - | |
| **Order VENERIDA** | | | |
| **Family VENERIDAE** | | | |
| *Pitar mediterraneus* (Aradas & Benoit, 1872) | 60 m | - | |
| *Venus verrucosa* Linnaeus, 1758 | - | - | |
| **Class CEPHALOPODA** | | | |
| **Order MYOPSIDA** | | | |
| **Family LOLIGINIDAE** | | | |
| *Loligo vulgaris* Lamarck, 1798 | 30 m | - | |
| **Order OCTOPODA** | | | |
| **Family OCTOPODIDAE** | | | |
| *Octopus vulgaris* Cuvier, 1797 complex | 2–30 m | - | |
| **Order SEPIIDA** | | | |
| **Family SEPIIDAE** | | | |
| *Sepia officinalis* Linnaeus, 1758 | 5–30 m | - | |
| **Class GASTROPODA** | | | |
| **Family CERITHIIDAE** | | | |
| *Bittium latreillii* (Payraudeau, 1826) | 10 m | - | |
| *Bittium reticulatum* (da Costa, 1778) | - | - | |
| *Cerithium lividulum* Risso, 1826 | 1 m | - | |
| *Cerithium renovatum* Monterosato, 1884 complex | - | - | |
| *Cerithium vulgatum* Bruguière, 1792 complex | 1 m | - | |

**Table 1.** *Cont.*

| Taxa | Depth | Phenology | New Record |
|---|---|---|---|
| **Family SILIQUARIIDAE** | | | |
| *Petalopoma elisabettae* Schiaparelli, 2002 | 40 m | - | |
| **Family LIMAPONTIIDAE** | | | |
| *Ercolania viridis* (A. Costa, 1866) [30] | 0 m | - | * |
| **Family OMALOGYRIDAE** | | | |
| *Ammonicera fischeriana* (Monterosato, 1869) | - | - | |
| *Omalogyra* cf. *simplex* (Costa O. G., 1861) | - | - | |
| **Family PATELLIDAE** | | | |
| *Patella caerulea* Linnaeus, 1758 | 0 m | - | |
| *Patella rustica* Linnaeus, 1758 complex | 0 m | - | |
| **Family PLAKOBRANCHIDAE** | | | |
| *Bosellia mimetica* Trinchese, 1891 [30] | 21 m | Eggs (VI) | |
| *Elysia timida* (Risso, 1818) [30] | 7 m | - | |
| *Elysia viridis* (Montagu, 1804) [30] | 10 m | - | |
| *Thuridilla hopei* (Vérany, 1853) [30] | 7 m | - | |
| **Family PYRAMIDELLIDAE** | | | |
| *Folinella excavata* (Phillippi, 1836) | - | - | |
| **Order APLYSIIDA** | | | |
| **Family APLYSIIDAE** | | | |
| *Aplysia depilans* Gmelin, 1791 [30] | 2 m | Rep. behav. (IX) | |
| *Aplysia fasciata* Poiret, 1789 [30] | 3 m | Rep. behav. (VII) | |
| *Aplysia punctata* (Cuvier, 1803) [30] | 10 m | Rep. behav. (II) | |
| **Order CEPHALASPIDEA** | | | |
| **Family AGLAJIDAE** | | | |
| *Aglaja tricolorata* Renier, 1807 [30] | 34 m | - | |
| **Family BULLIDAE** | | | |
| *Bulla striata* Bruguière, 1792 [30] | 10 m | - | |
| **Family HAMINOEIDAE** | | | |
| *Haminoea* ind Turton & Kingston [in Carrington], 1830 [30] | 1–40 m | - | |
| *Weinkauffia turgidula* (Forbes, 1844) [30] | 40 m | - | |
| **Order LEPETELLIDA** | | | |
| **Family FISSURELLIDAE** | | | |
| *Diodora graeca* (Linnaeus, 1758) | 1 m | - | |
| *Emarginula sicula* J.E. Gray, 1825 | 3 m | - | |
| **Family HALIOTIDAE** | | | |
| *Haliotis tuberculata* Linnaeus, 1758 | 2–10 m | - | |
| **Family SCISSURELLIDAE** | | | |
| *Scissurella costata* d'Orbigny, 1824 | - | - | |
| **Order LITTORINIMORPHA** | | | |
| **Family APORRHAIDAE** | | | |
| *Aporrhais pespelecani* (Linnaeus, 1758) | 3 m | - | |
| **Family CALYPTRAEIDAE** | | | |
| *Calyptraea chinensis* (Linnaeus, 1758) | 50 m | - | |
| *Crepidula moulinsii* Michaud, 1829 | 20 m | - | |
| **Family CARINARIIDAE** | | | |
| *Carinaria lamarckii* Blainville, 1817 | 3 m | - | |
| **Family CASSIDAE** | | | |
| *Semicassis undulata* (Gmelin, 1791) | - | - | |
| **Family CHARONIIDAE** | | | |
| *Charonia seguenzae* (Aradas & Benoit, 1872) | 11 m | - | |
| **Family CYMATIIDAE** | | | |
| *Monoplex corrugatus* (Lamarck, 1816) | 10 m | - | |
| **Family CYPRAEIDAE** | | | |
| *Luria lurida lurida* (Linnaeus, 1758) | 4 m | - | |
| *Naria spurca spurca* (Linnaeus, 1758) | 4 m | - | |
| **Family LITTORINIDAE** | | | |
| *Melarhaphe neritoides* (Linnaeus, 1758) | 0 m | Eggs (XI) | |
| **Family RISSOIDAE** | | | |

**Table 1.** *Cont.*

| Taxa | Depth | Phenology | New Record |
|---|---|---|---|
| *Alvania hirta* (Monterosato, 1884) | - | - | |
| *Alvania lineata* Risso, 1826 | - | - | |
| *Alvania mamillata* Risso, 1826 | - | - | |
| *Manzonia crassa* (Kanmacher, 1798) | - | - | |
| **Family RISSOINIDAE** | | | |
| *Rissoina bruguieri* (Payraudeau, 1826) | - | - | |
| **Family TONNIDAE** | | | |
| *Tonna galea* (Linnaeus, 1758) | 20–60 m | - | |
| **Family VELUTINIDAE** | | | |
| *Lamellaria perspicua* (Linnaeus, 1758) | 3 m | - | |
| **Family VERMETIDAE** | | | |
| *Dendropoma petraeum* (Monterosato, 1884) complex | - | - | |
| *Thylacodes arenarius* (Linnaeus, 1758) | 70 m | - | |
| **Order NEOGASTROPODA** | | | |
| **Family BUCCINIDAE** | | | |
| *Euthria cornea* (Linnaeus, 1758) | - | - | |
| **Family COLUMBELLIDAE** | | | |
| *Columbella rustica* (Linnaeus, 1758) | 1 m | - | |
| **Family CONIDAE** | | | |
| *Conus ventricosus* Gmelin, 1791 | 2 m | - | |
| **Family COSTELLARIIDAE** | | | |
| *Pusia tricolor* (Gmelin, 1791) | 4 m | - | |
| **Family FASCIOLARIIDAE** | | | |
| *Fusinus fioritae* Russo & Pagli, 2019 | 40 m | - | |
| *Tarantinaea lignaria* (Linnaeus, 1758) | 1 m | - | |
| **Family MITRIDAE** | | | |
| *Episcomitra zonata* (Marryat, 1818) | - | - | |
| *Isara cornea* (Lamarck, 1811) | - | - | |
| **Family MURICIDAE** | | | |
| *Bolinus brandaris* (Linnaeus, 1758) | 30 m | - | |
| *Hexaplex trunculus* (Linnaeus, 1758) complex | 1–50 m | Eggs (VI–VII) | |
| *Hirtomurex squamosus* (Bivona e Bernardi, 1838) | 50 m | - | |
| *Muricopsis cristata* (Brocchi, 1814) | 30 m | - | |
| *Ocinebrina* cf. *corallina/aegeensis* | 40 m | - | |
| *Stramonita haemastoma* (Linnaeus, 1767) | 0 m | - | |
| **Family NASSARIIDAE** | | | |
| *Tritia corniculum* (Olivi, 1792) complex | 1 m | - | |
| *Tritia incrassata* (Strøm, 1768) complex | 1 m | - | |
| *Tritia pellucida* (Risso, 1826) | 5 m | - | |
| **Family PISANIIDAE** | | | |
| *Aplus dorbignyi* (Payraudeau, 1826) complex | - | - | |
| *Pisania striata* (Gmelin, 1791) | 20 m | - | |
| **Order NUDIBRANCHIA** | | | |
| **Family AEOLIDIIDAE** | | | |
| *Berghia coerulescens* (Laurillard, 1832) [30] | 1 m | - | |
| *Spurilla neapolitana* (Delle Chiaje, 1841) [30] | 4 m | - | |
| **Family CALYCIDORIDIDAE** | | | |
| *Diaphorodoris papillata* Portmann & Sandmeier, 1960 [30] | 30 m | - | * |
| **Family CHROMODORIDIDAE** | | | |
| *Felimare picta* (Philippi, 1836) [30] | 8–18 m | - | |
| *Felimare tricolor* (Cantraine, 1835) [30] | 14 m | - | |
| *Felimare villafranca* (Risso, 1818) [30] | 10 m | - | |
| *Felimida krohni* (Vérany, 1846) [30] | 40 m | - | |
| *Felimida luteorosea* (Rapp, 1827) [30] | 40 m | - | |
| **Family DENDRODORIDIDAE** | | | |
| *Dendrodoris grandiflora* (Rapp, 1827) [30] | 4 m | - | |
| **Family DISCODORIDIDAE** | | | |
| *Peltodoris atromaculata* Bergh, 1880 [30] | 18–30 m | Eggs (VI) | |

**Table 1.** *Cont.*

| Taxa | Depth | Phenology | New Record |
|---|---|---|---|
| *Platydoris argo* (Linnaeus, 1767) [30] | 22 m | - | |
| **Family DORIDIDAE** | | | |
| *Doris ocelligera* (Bergh, 1881) [30] | 40 m | - | * |
| **Family DOTIDAE** | | | |
| *Doto acuta* Schmekel & Kress, 1977 [30] | 2 m | - | * |
| *Doto* cf. *koenneckeri* Lemche, 1976 [30] | 2 m | - | * |
| *Doto paulinae* Trinchese, 1881 [30] | 2 m | - | * |
| *Doto pygmaea* Bergh, 1871 [30] | 0 m | Eggs (VII) | * |
| **Family EUBRANCHIDAE** | | | |
| *Eubranchus exiguus* (Alder & Hancock, 1848) [30] | 0 m | - | * |
| **Family FACELINIDAE** | | | |
| *Cratena peregrina* (Gmelin, 1791) [30] | 21 - 25 m | - | |
| *Facelina annulicornis* (Chamisso & Eysenhardt, 1821) [30] | 4 m | - | * |
| **Family FIONIDAE** | | | |
| *Fiona pinnata* (Eschscholtz, 1831) [30] | 0 m | Eggs (V) | * |
| **Family FLABELLINIDAE** | | | |
| *Calmella cavolini* (Vérany, 1846) [30] | 1 m | - | |
| *Flabellina affinis* (Gmelin, 1791) [30] | 10 m | - | |
| **Family ONCHIDORIDIDAE** | | | |
| Onchidorididae ind Gray, 1827 [30] | 70 m | - | |
| **Family PHYLLIDIIDAE** | | | |
| *Phyllidia flava* Aradas, 1847 [30] | 23 m | - | |
| **Family POLYCERIDAE** | | | |
| *Kaloplocamus ramosus* (Cantraine, 1835) [30] | 70 m | - | |
| *Polycera quadrilineata* (O. F. Müller, 1776) [30] | 40 m | - | |
| **Family SAMLIDAE** | | | |
| *Luisella babai* (Schmekel, 1972) [30] | 22 m | - | |
| **Family TRINCHESIIDAE** | | | |
| *Trinchesia caerulea* (Montagu, 1804) [30] | 2 m | - | |
| **Family TRITONIIDAE** | | | |
| *Tritonia manicata* Deshayes, 1853 [30] | 1 m | - | |
| **Order PLEUROBRANCHIDA** | | | |
| **Family PLEUROBRANCHAEIDAE** | | | |
| *Pleurobranchaea meckeli* (Blainville, 1825) [30] | 34 m | - | |
| **Order RUNCINIDA** | | | |
| **Family RUNCINIDAE** | | | |
| *Runcina adriatica* T. E. Thompson, 1980 [30] | 20–39 m | - | * |
| *Runcina* cf. *brenkoae* T. E. Thompson, 1980 [30] | 2 m | - | * |
| *Runcina* cf. *ornata* (Quatrefages, 1844) [30] | 20 m | - | * |
| *Runcina ferruginea* Kress, 1977 [30] | 40 m | - | |
| **Order TROCHIDA** | | | |
| **Family CALLIOSTOMATIDAE** | | | |
| *Calliostoma conulus* (Linnaeus, 1758) | 4–60 m | - | |
| **Family COLLONIIDAE** | | | |
| *Homalopoma sanguineum* (Linnaeus, 1758) | - | - | |
| **Family TROCHIDAE** | | | |
| *Jujubinus exasperatus* (Pennant, 1777) complex | - | - | |
| *Jujubinus striatus* (Linnaeus, 1758) complex | 1 m | - | |
| *Phorcus turbinatus* (Born, 1778) | 0 m | - | |
| *Steromphala nebulosa* (Philippi, 1849) | 3 m | - | |
| **Family TURBINIDAE** | | | |
| *Bolma rugosa* (Linnaeus, 1767) | - | - | |
| **Order UMBRACULIDA** | | | |
| **Family UMBRACULIDAE** | | | |
| *Umbraculum umbraculum* ([Lightfoot], 1786) [30] | 1–3 m | - | |
| **Class POLYPLACOPHORA** | | | |
| **Order CHITONIDA** | | | |
| **Family ACANTHOCHITONIDAE** | | | |

**Table 1.** *Cont.*

| Taxa | Depth | Phenology | New Record |
|---|---|---|---|
| *Acanthochitona fascicularis* (Linnaeus, 1767) | 20 m | - | |
| **Family CALLOCHITONIDAE** | | | |
| *Callochiton septemvalvis* (Montagu, 1803) | 20 m | - | |
| **Family CHITONIDAE** | | | |
| *Rhyssoplax olivacea* (Spengler, 1797) | 10 m | - | |
| **Family LEPIDOCHITONIDAE** | | | |
| *Lepidochitona caprearum* (Scacchi, 1836) | 1 m | Eggs (V) | |
| | | | |
| **Phylum ANNELIDA** | | | |
| **Order MYZOSTOMIDA** | | | |
| **Family MYZOSTOMATIDAE** | | | |
| Myzostoma glabrum Graff, 1877 | 60 m | - | * |
| **Order SIPUNCULA** | | | |
| **Family PHASCOLOSOMATIDAE** | | | |
| *Phascolosoma (Phascolosoma) granulatum* Leuckart, 1828 | 0–1 m | - | |
| **Class CLITELLATA** | | | |
| **Order RHYNCHOBDELLIDA** | | | |
| **Family PISCICOLIDAE** | | | |
| *Branchellion torpedinis* Savigny, 1822 | - | - | * |
| *Pontobdella muricata* (Linnaeus, 1758) | - | - | |
| **Class POLYCHAETA** | | | |
| **Family OPHELIIDAE** | | | |
| *Armandia polyophthalma* Kükenthal, 1887 | 7 m | - | |
| **Family ORBINIIDAE** | | | |
| *Phylo foetida* (Claparède, 1868) | 18 m | - | |
| **Order AMPHINOMIDA** | | | |
| **Family AMPHINOMIDAE** | | | |
| *Hermodice carunculata* (Pallas, 1766) | 5–25 m | Eggs (VIII) | |
| **Order ECHIUROIDEA** | | | |
| **Family BONELLIIDAE** | | | |
| *Bonellia viridis* Rolando, 1822 | 5–15 m | - | |
| **Order EUNICIDA** | | | |
| **Family DORVILLEIDAE** | | | |
| *Dorvillea rubrovittata* (Grube, 1855) | 35 m | - | |
| **Family EUNICIDAE** | | | |
| *Leodice harassii* (Audouin & Milne Edwards, 1833) | 70 m | - | |
| *Lysidice collaris* Grube, 1870 | 70 m | - | |
| *Lysidice ninetta* Audouin & H Milne Edwards, 1833 | 15 m | - | |
| **Order PHYLLODOCIDA** | | | |
| **Family HESIONIDAE** | | | |
| *Psamathe fusca* Johnston, 1836 | 70 m | - | |
| **Family NEREIDIDAE** | | | |
| *Ceratonereis (Composetia) costae* (Grube, 1840) | 20 m | - | |
| **Family PHYLLODOCIDAE** | | | |
| Alciopini ind Ehlers, 1864 | 5 m | Occurrence (IV) | |
| **Family POLYNOIDAE** | | | |
| *Harmothoe* cf. *impar* (Johnston, 1839) | 35 m | - | |
| *Harmothoe pagenstecheri* Michaelsen, 1896 | 25 m | - | * |
| *Lepidonotus clava* (Montagu, 1808) | 70 m | - | |
| **Family SIGALIONIDAE** | | | |
| *Sigalion mathildae* Audouin & Milne Edwards in Cuvier, 1830 | 7 m | - | |
| **Family SYLLIDAE** | | | |
| *Paraehlersia ferrugina* (Langerhans, 1881) | 20 m | - | |
| *Pseudosyllis brevipennis* Grube, 1863 | 70 m | - | |
| **Order SABELLIDA** | | | |
| **Family SABELLIDAE** | | | |
| *Sabella spallanzanii* (Gmelin, 1791) | 3–20 m | - | |
| **Family SERPULIDAE** | | | |

**Table 1.** *Cont.*

| Taxa | Depth | Phenology | New Record |
|------|-------|-----------|------------|
| *Hydroides elegans* (Haswell, 1883) [nomen protectum] | 60 m | - | |
| *Hydroides pseudouncinata* Zibrowius, 1968 | 70 m | - | |
| *Protula* ind Risso, 1826 | 10–70 m | - | |
| *Spirobranchus triqueter* (Linnaeus, 1758) | 60 m | - | |
| *Spirorbis* ind Daudin, 1800 | 10 m | - | |
| *Vermiliopsis infundibulum* (Philippi, 1844) | 50–70 m | Eggs (VII) | |
| *Vermiliopsis labiata* (O. G. Costa, 1861) | 20 m | - | |
| *Vermiliopsis striaticeps* (Grube, 1862) | 18 m | - | |
| **Order TEREBELLIDA** | | | |
| **Family CIRRATULIDAE** | | | |
| *Cirriformia tentaculata* (Montagu, 1808) | 18 m | - | |
| *Dodecaceria concharum* Örsted, 1843 | 70 m | - | |
| **Family TEREBELLIDAE** | | | |
| *Streblosoma* ind M. Sars in G.O. Sars, 1872 | 35 m | - | |
| | | | |
| **Phylum NEMERTEA** | | | |
| **Class HOPLONEMERTEA** | | | |
| **Order POLYSTILIFERA** | | | |
| **Family DREPANOPHORIDAE** | | | |
| *Gibsonnemertes spectabilis* (Quatrefages, 1846) | 70 m | - | * |
| **Class PILIDIOPHORA** | | | |
| **Order HETERONEMERTEA** | | | |
| **Family LINEIDAE** | | | |
| *Notospermus geniculatus* (Delle Chiaje, 1828) | 4 m | - | |
| | | | |
| **Phylum BRYOZOA** | | | |
| **Class GYMNOLAEMATA** | | | |
| **Order CHEILOSTOMATIDA** | | | |
| **Family ADEONIDAE** | | | |
| *Adeonella pallasii* (Heller, 1867) | 20 m | - | |
| *Reptadeonella violacea* (Johnston, 1847) | 30 m | - | |
| **Family AETEIDAE** | | | |
| *Aetea anguina* (Linnaeus, 1758) | 0–5 m | - | |
| **Family BEANIIDAE** | | | |
| *Beania magellanica* (Busk, 1852) | 70 m | - | |
| **Family BITECTIPORIDAE** | | | |
| *Schizomavella* ind Canu & Bassler, 1917 | 70 m | - | |
| **Family BUGULIDAE** | | | |
| *Bugulina calathus* (Norman, 1868) | 50 m | - | |
| **Family CANDIDAE** | | | |
| *Caberea boryi* (Audouin, 1826) | 20–30 m | - | |
| **Family CELLARIIDAE** | | | |
| *Cellaria salicornioides* Lamouroux, 1816 | - | - | |
| **Family FLUSTRIDAE** | | | |
| *Chartella* ind Gray, 1848 | 70 m | - | * |
| **Family HIPPALIOSINIDAE** | | | |
| *Hippaliosina depressa* (Busk, 1854) | 10 m | - | |
| **Family MICROPORIDAE** | | | |
| *Calpensia nobilis* (Esper, 1796) | 5–50 m | - | |
| **Family MYRIAPORIDAE** | | | |
| *Myriapora truncata* (Pallas, 1766) | 10–70 m | - | |
| **Family PHIDOLOPORIDAE** | | | |
| *Reteporella couchii* (Hincks, 1878) | 25 m | - | |
| *Reteporella grimaldii* (Jullien, 1903) | 20 m | - | |
| *Schizoretepora serratimargo* (Hincks, 1886) | 10–25 m | - | |
| **Family SCHIZOPORELLIDAE** | | | |
| *Schizobrachiella sanguinea* (Norman, 1868) | 10 m | - | |
| **Order CTENOSTOMATIDA** | | | |

**Table 1.** *Cont.*

| Taxa | Depth | Phenology | New Record |
|---|---|---|---|
| **Family MIMOSELLIDAE** | | | |
| *Mimosella gracilis* Hincks, 1851 | 15 m | - | |
| **Family VESICULARIIDAE** | | | |
| *Amathia semiconvoluta* Lamouroux, 1824 | - | - | |
| **Family WALKERIIDAE** | | | |
| *Walkeria* ind Fleming, 1823 | 30 m | - | |
| **Class STENOLAEMATA** | | | |
| **Order CYCLOSTOMATIDA** | | | |
| **Family FRONDIPORIDAE** | | | |
| *Frondipora verrucosa* (Lamouroux, 1821) | 60 m | - | |
| | | | |
| **Phylum BRACHIOPODA** | | | |
| **Class RHYNCHONELLATA** | | | |
| **Order TEREBRATULIDA** | | | |
| **Family MEGATHYRIDIDAE** | | | |
| *Argyrotheca cuneata* (Risso, 1826) | 70 m | - | |

## 4. Discussion

Biodiversity is currently a central theme of global environmental policies and conservation strategies (e.g., EU's Marine Strategy Framework Directive). However, a sound understanding of biodiversity should be considered at the foundation of any conservation policy. Despite that, our 18-month study showed that the knowledge of Mediterranean biodiversity can still be significantly improved.

This study provides the first species inventory (marine organisms and coastal flora) of the southeastern Salento coastal area and enriches the current basic knowledge information on the biodiversity of the Ionian and the central Mediterranean Sea. Overall, 696 taxa were identified, among which Mollusca, Porifera and Chordata were the most represented, accounting for more than half of the full species inventory. Despite that, the groups with the highest percentages of new records were Polycladida (Platyhelminthes), Nemertea, and Thaliacea, with the relative percentage of new records of 80%, 50% and 33%, respectively.

For some groups, these results reflect the knowledge gaps in their taxonomy and ecology. Some taxa are poorly studied and lack Mediterranean experts, leading to inaccurate estimates of species occurrence. Moreover, the scientific community nowadays tends to overlook descriptive research and biodiversity records [63]. For example, for polyclad platyhelminths and nemerteans, reference monographs for the Mediterranean Sea date back to the end of the 19th century, when they were studied in the Gulf of Naples [64,65]. Similarly, for the phylum Ctenophora, most of the records from Italy come from the Gulf of Naples and the Strait of Messina, the latter being the only location where most ctenophores have been reported in the Ionian Sea (Figure 6b) [66]. Some groups, like anthozoans, have been poorly surveyed in the Ionian Sea, which explains why one tenth of the species found by this study represented new records for the area [67]. For other groups (i.e., Heterobranchia and Porifera), the high proportion of new records and new species (sponges) likely reflects the lack of expert work in the area and the difficulty of taxonomic identification (for an in-depth discussion on these two groups, see [30,33]).

Comparing our species inventory with other works in the central and eastern Mediterranean Sea, the results vary by taxonomic group. However, species inventories at a very small spatial scale, such as ours, are rare, so comparisons are made with caution. Regarding sponges, we reported 112 taxa, while Evcen & Cinar [68] reported 116 taxa of sponges for the whole coast of Turkey, and Voultsiadou et al. [69] reported 81 taxa for the Aegean Sea. The fact that the coastlines of these areas are about two orders of magnitude longer than our study area suggests that the southeastern Salento has a relatively high diversity of sponges, as suggested by Sarà [26]. Regarding hydrozoans, Morri et al. [70] reported 38 taxa for Lebanon, raising the total number of hydrozoans in the Levant Sea to 70.

Morri & Bianchi [71] reported 31 taxa for the Aegean Islands of Kos, raising to 67 the number of hydroids known from the Aegean Sea and nearby areas. Also in this case, considering that these areas are several orders of magnitude larger than southeastern Salento, our finding of 48 taxa represents a relatively high number, suggesting a high diversity in the class Hydrozoa. Regarding anthozoans, Vafidis et al. [72] found 21 taxa belonging to the orders Actiniaria, Corallimorpharja, and Scleractinia in the northern Aegean Sea, while we reported 16 of them, which could be comparable considering the smaller area investigated by our study. Also for molluscs, our species list is comparable to the one realised by Giacobbe & Renda [73] for Capo D'Armi, Sicily. They reported 133 taxa, compared to 144 taxa found in our study. The higher number of our species could be explained by the broader area and depth range of our study. Finally, regarding bony fishes, our list only reports 89 species, while Al-Hassan & El-Silini [74] reported 201 species for the coast of Benghazi, Libya, and Saad [75] reported 224 species for the coast of Syria. These large differences can be related to the larger survey areas in Al-Hassan & El-Silini [74], and to the higher sampling effort focused specifically on this group of organisms.

Comparing our species number with the total number of species in the Mediterranean Sea [7], the groups with a proportionally higher number of species were Anthozoa (19%), Porifera and Echinodermata (16%), Osteichthyes (~14%) and Decapoda (13%) (Table 2). Similarly, compared to the checklist for Italian waters, we recorded 24% of Anthozoa, 22% of Porifera, 20% of Echinodermata and Osteichthyes and 17% of Decapoda. Finally, compared to the checklist of the Italian Ionian Sea, we recorded 78% of Anthozoa, 58% of Porifera, 31% of Echinodermata, 29% of Hydrozoa and 25% of Osteichthyes (Table 2). However, it is important to emphasize that the known distribution for many marine taxa is directly related to the distribution of taxonomists and does not reflect the true distribution of the species [76,77]. Therefore, caution is needed in interpreting the numerical comparisons made in this and the previous paragraph.

**Table 2.** Comparison of the results of this study with the total number of taxa in the Mediterranean Sea [7], Italian waters [60,61] and Italian Ionian Sea [60,61], for selected taxonomic groups. Numbers represent the number of taxa, while the percentage of taxa found by this study out of the total number of species known for the area is given in brackets. * this figure also includes Chondrichthyes.

|  | Our Study | Mediterranean Sea | Italian Waters | Italian Ionian Sea |
|---|---|---|---|---|
| Porifera | 112 | 681 (16%) | 509 (22%) | 193 (58%) |
| Gastropoda | 107 | 1564 (7%) | 1155 (9%) | 763 (14%) |
| Osteichthyes | 89 | * 650 (14%) | 442 (20%) | 351 (25%) |
| Macrophyte | 88 | 1131 (8%) | 949 (9%) | 736 (12%) |
| Hydrozoa | 49 | 457 (11%) | 346 (14%) | 171 (29%) |
| Decapoda | 49 | 383 (13%) | 293 (17%) | 212 (23%) |
| Annelida | 33 | 1172 (3%) | 951 (3%) | 577 (6%) |
| Anthozoa | 31 | 164 (19%) | 128 (24%) | 40 (78%) |
| Bivalvia | 29 | 400 (7%) | 340 (9%) | 227 (13%) |
| Echinodermata | 24 | 154 (16%) | 120 (20%) | 77 (31%) |

Out of a total of 697 taxa, we reported only 9 NIS. Considering other areas of the Apulia region, this is a considerable low number of NIS [78]. NIS are often opportunistic species that proliferate in heavily impacted ecosystems [79]. The low numbers of NIS we reported in this study may reflect the overall good environmental conditions and lack of major anthropogenic disturbances in the Tricase coastal area.

Our results also depend on sampling methodologies. For instance, we reported only 32 species of annelids, while there are 152 taxa of hard-substrate polychaetes reported in the Otranto channel [36]. This was probably the result of not having performed ad hoc sampling, e.g., [80,81]. Importantly, this work is not intended to be an exhaustive inventory of species in the area, but a baseline. Future in-depth studies on single groups of fauna and flora will be necessary and will likely disclose further diversity of this stretch of coast.

Besides taxonomic groups, the highest proportion of new records was found within the group of gelatinous zooplankton (i.e., scyphozoan and hydrozoan jellyfish, siphonophores, thaliaceans, and ctenophores). Gelatinous plankton is often the dominant macrozooplankton of oceanic systems, and an important component of marine ecosystems, with particular significance for fisheries management and the tourism industry [82,83]. Despite its importance, gelatinous plankton remains one of the less studied and understood marine groups [84]. The reason lies mainly in the difficulty of sampling animals with fragile gelatinous bodies and in their irregular occurrence as adult forms, which makes it difficult to plan sampling campaigns [85,86]. Marine stations, such as the MARE Outpost, offer a significant advantage for the study of these organisms since they allow continuous surveillance of the coastal area and easy sampling by SCUBA and free diving. In addition, citizen science could also offer substantial help in the study of gelatinous plankton, as it can enable monitoring programs on large geographic and temporal scales, while increasing biodiversity awareness among the general public, e.g., [86–91].

Other taxonomic groups appeared to be relatively well studied, with no new records despite the high number of taxa reported. Among these, we found fishes (Osteichthyes), macroalgae, shelled molluscs, decapods (Crustacea), and echinoderms. Fishes are particularly well studied in the Mediterranean Sea, also because of their commercial interest [7,92]. Shelled molluscs are well-known partly due to contributions from amateur shell collectors, who far outnumber professional researchers [93].

Increased knowledge of species distribution and phenology is critical to understand the effects of climate change and human actions on ecosystems and assessing good environmental status, as required by the Marine Strategy Framework Directive. Therefore, long-term biodiversity monitoring and observation are needed to improve our knowledge of biodiversity changes. There is also a need for new taxonomists who can identify and research marine organisms [94]. Inspired by the seminal work of Salvatore Lo Bianco [95] in the Gulf of Naples, this work is an important addition to local and regional biodiversity knowledge and a baseline for future biodiversity monitoring in the Ionian and Mediterranean Sea.

**Supplementary Materials:** The following supporting information can be downloaded at: https://www.mdpi.com/article/10.3390/d14110904/s1, Table S1: Complete list of marine species and coastal plants found during the project Biodiversity MARE Tricase with the indication of the depth or depth range at which the taxa were recorded, the number of specimens sampled and recorded, phenology, whether the taxa is a non-indigenous species, whether the taxa is a new record for the area, and the reference for records already published in previous publications by the same authors [30,33]; Table S2: Complete list of marine specimens recorded and identified during the project Biodiversity MARE Tricase with relative metadata and notes.

**Author Contributions:** Conceptualization: V.M., F.S., S.P. and F.B. (Ferdinando Boero); Supervision: F.B. (Ferdinando Boero) and S.P.; Investigation: V.M. and F.S.; Taxonomic identification: F.C., D.D.F., C.G., F.R., M.B. (Marco Bertolino), G.C., J.L., M.B. (Marzia Bo), F.B. (Federico Betti), C.F., A.G., F.T., L.N., P.M. and S.A.; Writing—Original Draft Preparation, V.M., F.S.; Writing—Review & Editing, all the authors. All authors have read and agreed to the published version of the manuscript.

**Funding:** The project "Biodiversity MARE Tricase" was partially funded by the PADI Foundation Grant 2017 (28815) and by the Italian Zoological Union (UZI), and the Scientific Committee for the Italian Fauna (CSFI) through a prize for the best poster on the Italian Fauna at the National Joint Conference of the Italian Society of Ecology (SItE), the Italian Zoological Union (UZI), and the Italian Society of Biogeography (SIB).

**Institutional Review Board Statement:** Not applicable.

**Data Availability Statement:** The data presented in this study are available in the Supplementary Materials, Tables S1 and S2.

**Acknowledgments:** This work was possible thanks to the establishment of Avamposto MARE. A special thanks to Antonio Errico and Salvatore Baglivo ("Magna Grecia Mare" Association), Antonio

Coppola (Former Mayor of Tricase), Maurizio Raeli and Biagio Di Terlizzi (CIHEAM Bari). We would like to thank the community of Tricase, starting from the "Deep Water Divers" (Salvatore Bortone and Anna Rita Chiuri), "Ittiturismo Anime Sante" (Daniele, Francesco, and Rocco Cazzato), "Masseria Nonno Tore (Irene Vaglio and Carlo Panico)", Alessandro, Marco and Rocco Ferrarese, and Lucio Cazzato and Giulio, for their precious help. We would also thank the trainees, Emanuele Astoricchio (Italy), Laura Basconi (Italy), Anna Berti (Italy), Jessica de Felice (Italy), Jade Sourisse (France), Olivier Vangheluwe (France), and the work-linked training students Maria Assunta Corciulo, Giacomo Luzzati and Noemi Marra. Thanks also to Egidio Trainito for his scientific support.

**Conflicts of Interest:** The authors declare no conflict of interest.

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
