# Peer review of "Project “Biodiversity MARE Tricase”: A Species Inventory of the Coastal Area of Southeastern Salento (Ionian Sea, Italy)"

_diversity, doi:10.3390/d14110904_

Round 1

Reviewer 1 Report

Naturalist inventories are the basis of ecology and the protection of the marine environment. However, they are increasingly rare; This work has the merit of returning to this fundamental and of recalling that the first step in management is knowledge. In this sense, this publication is of interest beyond the actual inventory of species. The publication also raises the need for specialists in species determination. Remarks : - line 75: add coastal urbanization pressure - specify in the text "equipment and methods" the observation methods for diving (transect, circular point) - what was the most used sampling technique? Congratulations on the identification of new species and this great inventory work.

Author Response

Naturalist inventories are the basis of ecology and the protection of the marine environment. However, they are increasingly rare; This work has the merit of returning to this fundamental and of recalling that the first step in management is knowledge. In this sense, this publication is of interest beyond the actual inventory of species. The publication also raises the need for specialists in species determination.

Remark 1: line 75: add coastal urbanization pressure.

Response: We have now added "coastal urbanization pressure" to the sentence in line 74.

Remark 2: specify in the text "equipment and methods" the observation methods for diving (transect, circular point) - what was the most used sampling technique?

Response: We have now improved the M&M section about sampling methods (lines 131-154), and we have added a paragraph in the results about most used sampling techniques and a breakdown of the specimens collected with these (lines 285-294)

Remark 3: Congratulations on the identification of new species and this great inventory work.

Response: Thank you for appreciating our work.

Reviewer 2 Report

General comments

The manuscript provides interesting information on the biodiversity of an area of high biogeographical value such as the Salento Peninsula between the Ionian and Adriatic seas and enriches the current basic knowledge information of these areas. The sure classification of the species is guaranteed by the taxonomists who participate in the publication.

However, given the important review work carried out, it would have been convenient to include all the taxa cited in the study area (in Table S1) and that the authors have not found in their study

Specific comments In the text, why aren't species/genus in italics?

It would be more scientific in table 1 to order by kingdom/phyla to follow a phylogenetic order, starting with the bacteria and ending with the chordates (not alphabetically).

Table 2: Does not need to be included, since the species that are new citations are indicated with an asterisk in Table 1.

Pag. 11, range 260: (Fig. XX)… is (Table S2)

Author Response

General comments

The manuscript provides interesting information on the biodiversity of an area of high biogeographical value such as the Salento Peninsula between the Ionian and Adriatic seas and enriches the current basic knowledge information of these areas. The sure classification of the species is guaranteed by the taxonomists who participate in the publication.

Remark 1: However, given the important review work carried out, it would have been convenient to include all the taxa cited in the study area (in Table S1) and that the authors have not found in their study

Response: We thank the reviewer for the comment. This is a really good point and we have thought a lot about doing this in the past. However, we cannot be sure that the species we have not found are still present in the Tricase Area. Indeed, most bibliographic records for this area come from relatively old studies (e.g., Sarà 1969; Parenzan 1983). This also given the marked biological changes that the Mediterranean Sea has undergone during the last 20-30 years. For this reason, we prefer only to report the species/taxa that we found during our study. However, for completeness, we now report other available species lists for the area in the introduction (lines 102-105)  

Remark 2: In the text, why aren't species/genus in italics?

Response: Sorry about that, these were originally in italics, but they must have been changed during the initial editorial process. We have now formatted correctly species and genera along the text.

Remark 3: It would be more scientific in table 1 to order by kingdom/phyla to follow a phylogenetic order, starting with the bacteria and ending with the chordates (not alphabetically).

Response: The kingdoms and phyla have been ordered following phylogenetic order as suggested. We used the reference suggested by Reviewer 3 for the phylogenetic order (Giribet & Edgecombe, 2020).

Remark 4: Table 2: Does not need to be included, since the species that are new citations are indicated with an asterisk in Table 1.

Response: We thank the reviewer for the suggestion, Table 2 has now been removed.

Remark 5: Pag. 11, range 260: (Fig. XX)… is (Table S2)

Response: "(Fig. XX)" has been removed from the text.

Reviewer 3 Report

This an interesting paper on the total marine biodiversity at species level (mostly) of a small sector of the Mediterranean Sea. Therefore, it represents a baseline potentially useful to monitor future alterations in the marine ecosystems. The study summarizes the current knowledge, including groups seldom treated in the literature, handed by recognized specialists in the field. The literature cites many  relevant  papers of general significance, plus obviously many related to the specific situation considered by the team. Perhaps, AA slightly indulge in citing research involving the participants to this article, ignoring other quotable information, but this choice at the end does not undermine the net result of this study.

Minor points below:

Abstract:, line 37, add 'marine' before 'basins'; add 'pollution,' between 'resources' and 'climate'

Line 122, 'consisting of limestone', add 'bedrock'

Line 160 and following, Under Results, Genera and species in italics

References:

line 15, 'Halecium' in italics

line 25, 'Rhabdopleura recondita' in italics

Fig. 4, Genera and species in italics

Table 1, Patella rustica Linnaeus, 1758 complex 1 m : uncorrect, it is a species typically inhabiting the upper fringe of the intertidal, higher than other limpets; correct at least 0-1m (never seen so deep but..)

Carinaria lamarckii Blainville, 1817 3 m, holoplankton, it lives only at 3 m? adjust 

Author Response

Remark 1: Perhaps, AA slightly indulge in citing research involving the participants to this article, ignoring other quotable information, but this choice at the end does not undermine the net result of this study.

Response: We have now expanded the discussion and introduction citing and discussing similar works is other areas of the central and eastern Mediterranean Sea.

Minor points below:

Remark 2: Abstract:, line 37, add 'marine' before 'basins'; add 'pollution,'

between 'resources' and 'climate'

Response: We thank the reviewer for the useful suggestions. We have addressed the comment

Remark 3: Line 122, 'consisting of limestone', add 'bedrock'

Response: We have added bedrock

Remark 4: Line 160 and following, Under Results, Genera and species in italics

line 15, 'Halecium' in italics

line 25, 'Rhabdopleura recondita' in italics

Fig. 4, Genera and species in italics

Response: Species and genera were originally in italics, but they must have been changed during the initial editorial process. We have now formatted them correctly along the text.

Remark 5: Table 1, Patella rustica Linnaeus, 1758 complex 1 m : uncorrect, it is a

species typically inhabiting the upper fringe of the intertidal, higher

than other limpets; correct at least 0-1m (never seen so deep but..)

Response: We thank the reviewer for noticing the error, we have now replaced “1m” with “0m”

Remark 6: Carinaria lamarckii Blainville, 1817 3 m, holoplankton, it lives only at

3 m? adjust

Response: We only reported depth at which we found the organisms. In the case of Carinaria lamarckii we only found one specimen at 3 m of depth. We have now clarified what “depth” means in the caption of Table 1 and Table S1. (line 619).

Reviewer 4 Report

This manuscript deals on a taxonomic study of the species collected in the Salento region (Ionian Sea). Undoubtedly, this kind of inventories contributes to the knowledge of the flora and fauna of the corresponding sampled areas and, therefore, they are interesting for future faunistic studies and deserve to be published. However, in this specific case the results are based on a poorly elaborated list that deserves an exhaustive review.

Some recommendations to improve the content of the manuscript:
-First of all, a better ordered list using phylogenetic criteria should be presented [see Giribet & Edgecombe (2020). The invertebrate tree of life. Princeton University Press].
-Table 2 should be deleted and the new records for the area be included in a column of a single Table 1.
-The presence of the species in other Mediterranean regions, their abundance, etc. can be included in columns of the same table, to be able to comment in the text: their importance in the sampled communities; if there are no Mediterranean NIS (non-indigenous species), to discuss why; etc. The style of the current supplementary Table S1 can be followed.
-The study does not include a list of samples studied in the Materials and Methods section. The number of samples and their data are unknown. It would be interesting to add it even if it was obviously long, at least as a supplementary table.

-Some groups are very well represented (for example, Mollusca and Porifera) and others are surprisingly almost absent (for example, Amphipod and Isopod Crustaceans). Although possible explanations are indicated in the text (lines 317-322), it does not appear that the sampling (lines 133-138) is biased and there should not be these differences. Please, provide a more detailed argument.
-It would be interesting to compare the results with similar inventories from the eastern and western Mediterranean. This would also improve the Introduction section because it would give a more inclusive approach to the study in the context of the whole Mediterranean Sea.
-The diversity in relation to the total of Mediterranean species should be discussed, at least for the most well studied taxa (for example, Fishes, Mollusca, Porifera, Polychaeta, Crustacea Decapoda).
-Regarding the number of new species for science, there are only two. It seems a very low number for 697 identified taxa. Please, discuss why.

-In the Discussion section (lines 303-322): then... it would be better to include in this study only the most well studied taxa and delete additional groups?

Other minor changes:
-Line 152: reference 47 needs to be added.
-From line 155, the numbering of the citations must be revised, now do not correspond to the indicated references.
-Line 365: Table S3 does not exist.

-Supplementary Tables: Please, check the specific name column of Coastal Plants in Table S1, the names are repeated many times. Also, if the content of Table S2 is included in Table S1, it should be deleted.

Therefore, I believe that the manuscript must include the aspects indicated above and it is necessary to submit a second improved version.

Author Response

This manuscript deals on a taxonomic study of the species collected in the Salento region (Ionian Sea). Undoubtedly, this kind of inventories contributes to the knowledge of the flora and fauna of the corresponding sampled areas and, therefore, they are interesting for future faunistic studies and deserve to be published. However, in this specific case the results are based on a poorly elaborated list that deserves an exhaustive review.

Some recommendations to improve the content of the manuscript:

Remark 1: First of all, a better ordered list using phylogenetic criteria should be presented [see Giribet & Edgecombe (2020). The invertebrate tree of life. Princeton University Press].

Response: The phyla have been ordered following phylogenetic order as suggested using Giribet & Edgecombe (2020) for invertebrates, Goh et al. (2019) for Plantae, and Cavalier-Smith (2018) for Chromista.

Remark 2: Table 2 should be deleted and the new records for the area be included in a column of a single Table 1.

Response: We thank the reviewer for the suggestion. Table 2 has been eliminated and the new records are now listed in a separate column of Table 1

Remark 3: The presence of the species in other Mediterranean regions, their abundance, etc. can be included in columns of the same table, to be able to comment in the text: their importance in the sampled communities;

Response: We thank the reviewer for suggesting the addition of Mediterranean distribution and abundance to the taxa table. We have now added the number of specimens collected for each species, and, for limited species, the indication if these were common, uncommon or rare. Indeed, our sampling methods were designed to obtain observational and qualitative data. Therefore, we can't provide reliable quantitative information or define the ecological functions of the species listed in this study. We are only able to provide a naturalistic description of the area with the most common species, as we have done in the result section 3.1, from line 174.

Regarding reporting the presence of the species in other Mediterranean regions, this present species list is not limited to one taxonomic group, but includes many groups, and it has been carried out to specifically capture the biodiversity of Tricase coastal area. Adding the information on the Mediterranean distribution of all the taxa we reported would not fall within the aims of our study. Moreover, we don’t have the resources to do that for 697 taxa.

Remark 4: if there are no Mediterranean NIS (non-indigenous species), to discuss why; etc. The style of the current supplementary Table S1 can be followed.

Response: We thank the reviewer for raising the important aspect of NIS. In this regard, we have now added information about the occurrence of NIS in Table S1, in the results (lines 272-276) and discussion (lines 402-406) sections.

Remark 5: The study does not include a list of samples studied in the Materials and Methods section. The number of samples and their data are unknown. It would be interesting to add it even if it was obviously long, at least as a supplementary table.

Response: We thank the reviewer for this comment. Originally we did not provide additional metadata of the species list, given the large number of specimens collected. But we recognise this is a useful addition, and we have now included the list of the specimens collected on which we compiled the species list (Table S2). This includes the date of sampling, specific site, coordinates, depth, the temperature of the water, sampling method, substrate, phenology, and additional notes. In addition we now provide a breakdown of these additional pieces of information in the results (lines 285-297). Please note: we have removed from the list all the specimens that could not be identified, therefore the number of specimens provided in the results (1032, line 210) does not match the number of specimens in the specimens table S2 (878).

Remark 6: Some groups are very well represented (for example, Mollusca and Porifera) and others are surprisingly almost absent (for example, Amphipod and Isopod Crustaceans). Although possible explanations are indicated in the text (lines 317-322), it does not appear that the sampling (lines 133-138) is biased and there should not be these differences. Please, provide a more detailed argument.

Response: We thank the reviewer for this comment, that indeed clarifies an important aspect of the present study. Our sampling methodology was mainly focused on macro-fauna and flora, so we excluded taxonomic groups (e.g. meiofauna) and this aspect is already discussed (lines 407-409). In addition, the absence of certain taxonomic groups in the present list is also due to the lack of expertise. For example, Amphipoda and Isopoda were excluded from the list because we didn’t have a taxonomist for these groups in our team. Initially, samples of Amphipoda and Isopoda were sent for taxonomic identification to a collaborator, but this expert not only didn’t identify the specimens but also never sent us back the samples, preventing the inclusion of those groups in the current work. This aspect is now discussed in lines 410-411: "In addition, some groups (i.e., Amphipoda and Isopoda) were not included in the list because of the lack of taxonomic expertise in our team".

Remark 7: It would be interesting to compare the results with similar inventories from the eastern and western Mediterranean. This would also improve the Introduction section because it would give a more inclusive approach to the study in the context of the whole Mediterranean Sea.

Response: We have now added a paragraph in the discussion comparing our results of the best-studied groups with other available species lists for the central and eastern Mediterranean Sea (Lines 357-380).

Remark 8: The diversity in relation to the total of Mediterranean species should be discussed, at least for the most well studied taxa (for example, Fishes, Mollusca, Porifera, Polychaeta, Crustacea Decapoda).

Response: We have now added a paragraph in the discussion, comparing the numbers we found with regional species inventories (Mediterranean Sea, Italian waters and Italian Ionian Sea) (Lines 395-401) We have also added a table (Tab. 2, line 386) that provides a breakdown of this comparison for the main groups.

Remark 9: Regarding the number of new species for science, there are only two. It seems a very low number for 697 identified taxa. Please, discuss why.

Response: The description of new species was beyond the scope of our work. In the case of sponges, the taxonomists that collaborated with us were willing to carry out the description of new species, and this resulted in the description of two new species. For completeness, this information was reported in the manuscript. For other taxonomic groups, the collaboration only involved the identification of already described species.

Remark 10: In the Discussion section (lines 303-322): then... it would be better to include in this study only the most well studied taxa and delete additional groups?

Response: We believe that excluding these groups, even if not studied in depth would result in an unnecessary loss of important data. Indeed, some of these species belonging to these groups represent new records. However, also for records that are not new findings for the Ionian Sea, our records increase the resolution of our knowledge of these species in space and time, and provide important data points.

Remark 11: Line 152: reference 47 needs to be added.

Response: We have now corrected the reference list.

Remark 12: From line 155, the numbering of the citations must be revised, now do not correspond to the indicated references.

Response: We have now corrected the reference list.

Remark 13: Line 365: Table S3 does not exist.

Response: the text has been modified: "The data presented in this study are available in the supplementary materials, Table S1".

Remark 14: Supplementary Tables: Please, check the specific name column of Coastal Plants in Table S1, the names are repeated many times.

Response: Supplementary tables have been checked and there are no repeated names: the nomenclature of the Coastal Plants follows Pignatti S., 2017-2019 (Flora d’Italia, 4 voll., Edagricole, Bologna).

Remark 15: Also, if the content of Table S2 is included in Table S1, it should be deleted.

Response: Table S2 has been removed as suggested.

Round 2

Reviewer 4 Report

I have tried to check the authors' responses to the comments and sometimes it is difficult to find the correction in the text because the numbering of the lines is not consecutive, for example lines 187-213 and 268-292 are missing, although this does not affect the content to be published.
The Ms is now much improved, reasonably incorporating the suggested changes and, after amending some remaining details, I believe it will be ready to publish. For example, note that in the response to comment 6 you said "In addition, some groups (i.e., Amphipoda and Isopoda) were not included..." and, instead, in the list of species the Amphipoda and Isopoda are still included (pp. 35 and 37). I think it would be nice to prepare and review a final version of the text without additional notes.
Finally, my congratulations to the authors for this important contribution.

Author Response

We thank the reviewer a lot for the useful comments that have substantially improved the manuscript. We removed the additional note as suggested by the reviewer.